# NuRD subunit CHD4 regulates super-enhancer accessibility in rhabdomyosarcoma and represents a general tumor dependency

Joana G Marques[1], Berkley E Gryder[2], Blaz Pavlovic[1], Yeonjoo Chung[1], Quy A Ngo[1], Fabian Frommelt[3], Matthias Gstaiger[3], Young Song[2], Katharina Benischke[1], Dominik Laubscher[1], Marco Wachtel[1], Javed Khan[2]*, Beat W Schäfer[1]*

[1]Department of Oncology and Children's Research Center, University Children's Hospital, Zurich, Switzerland; [2]Oncogenomics Section, Genetics Branch, National Cancer Institute, National Institutes of Health, Bethesda, United States; [3]Department of Biology, Institute of Molecular Systems Biology, ETH Zurich, Zurich, Switzerland

**Abstract** The NuRD complex subunit CHD4 is essential for fusion-positive rhabdomyosarcoma (FP-RMS) survival, but the mechanisms underlying this dependency are not understood. Here, a NuRD-specific CRISPR screen demonstrates that FP-RMS is particularly sensitive to CHD4 amongst the NuRD members. Mechanistically, NuRD complex containing CHD4 localizes to super-enhancers where CHD4 generates a chromatin architecture permissive for the binding of the tumor driver and fusion protein PAX3-FOXO1, allowing downstream transcription of its oncogenic program. Moreover, CHD4 depletion removes HDAC2 from the chromatin, leading to an increase and spread of histone acetylation, and prevents the positioning of RNA Polymerase 2 at promoters impeding transcription initiation. Strikingly, analysis of genome-wide cancer dependency databases identifies CHD4 as a general cancer vulnerability. Our findings describe CHD4, a classically defined repressor, as positive regulator of transcription and super-enhancer accessibility as well as establish this remodeler as an unexpected broad tumor susceptibility and promising drug target for cancer therapy.

*For correspondence: khanjav@mail.nih.gov (JK); beat.schaefer@kispi.uzh.ch (BWSä)

Competing interests: The authors declare that no competing interests exist.

## Introduction

Chromatin remodelers regulate gene expression by controlling DNA accessibility to the transcriptional machinery at regulatory sites (*Clapier and Cairns, 2009*). The remodeling process is conducted by SNF2-like ATPases, usually working as subunits of multiprotein complexes, which use energy drawn from ATP hydrolysis to assemble, evict and move nucleosomes or exchange histone variants (*Clapier et al., 2017*). Besides gene expression regulation, these ATP-dependent remodelers are implicated in various fundamental cellular processes such as genome replication and DNA damage repair as well as cancer development (*Becker and Workman, 2013*; *Medina et al., 2008*; *Mills, 2017*). However, the role of chromatin remodeling in tumorigenesis is still poorly understood and few remodelers have been considered as possible drug targets for cancer therapy (*Mayes et al., 2014*).

The nucleosome remodeling and histone deacetylase (NuRD) complex is a highly conserved and ubiquitously expressed multisubunit complex (*Torchy et al., 2015*) which plays an essential role during normal development as well as in tumorigenesis (*Lai and Wade, 2011*). This complex combines

both chromatin remodeling (carried out by CHD3/4/5) and histone deacetylase (attributed to HDAC1/2) activity. Besides the catalytic subunits, the NuRD incorporates several non-enzymatic components including MBD2/3 (methyl-CpG-binding domain), RBBP4/7 (retinoblastoma-binding proteins), MTA1/2/3 (metastasis-associated proteins) and GATAD2A/B (GATA zinc finger domain containing proteins) (*Allen et al., 2013*; *Kolla et al., 2015*). In some instances, LSD1 (histone demethylase 1) (*Wang et al., 2009*) and CDK2AP1 (cyclin-dependent kinase 2 associated protein 1) (*Spruijt et al., 2010*) have been described as additional NuRD complex components. The NuRD subunits assemble in a combinatorial fashion and variations in the complex composition may reflect changes in its activity (*Bowen et al., 2004*). Currently, structural studies suggest that this complex is composed of two HDACs, two MTAs, four RBBPs and one MBD, GATAD2 and CHD subunits (*Torchy et al., 2015*; *Torrado et al., 2017*). The NuRD, partially due to its deacetylase activity, was originally defined as a transcription repressor (*Xue et al., 1998*), however there is increasing evidence suggesting that it might mediate both positive and negative regulation of gene expression (*Bornelöv et al., 2018*; *Günther et al., 2013*; *Hosokawa et al., 2013*; *Miccio et al., 2010*).

Fusion-positive rhabdomyosarcoma (FP-RMS) is a rare pediatric sarcoma with a low mutational burden that exhibits features of skeletal myogenesis (*Shern et al., 2014*). Its tumorigenesis is associated with the presence of chromosomal translocations which result in the expression of fusion oncogenic transcription factors. PAX3-FOXO1, the product of the most common chromosomal translocation observed, t(2;13)(q35;q14) (*De Giovanni et al., 2009*; *Shern et al., 2014*), drives tumor development by binding to enhancers and super-enhancers to activate an aberrant gene expression signature (*Cao et al., 2010*; *Gryder et al., 2017*; *Khan et al., 1999*). Since FP-RMS is dependent on its unique fusion genes (*Bernasconi et al., 1996*), efforts to further understand how PAX3-FOXO1 regulates gene expression and therapeutic strategies aiming to interfere with the fusion protein activity have been pursued. Recently, we demonstrated that the NuRD complex subunit and SNF2-like ATPase CHD4 (chromodomain-helicase-DNA-binding protein 4) is essential for FP-RMS survival and co-regulates the expression of a subset of PAX3-FOXO1 target genes (*Böhm et al., 2016*). However, the exact mechanisms by which CHD4 exerts this effect and whether the NuRD complex as such is involved are not yet clear. Hence, the goal of this study was to explore in detail the dependency of FP-RMS to the chromatin remodeler CHD4, to understand how it controls the oncogenic signature of PAX3-FOXO1 and to investigate the interplay between CHD4 and the NuRD complex in the context of this malignancy. Here, we describe a new role for CHD4 as a regulator of super-enhancer accessibility and super-enhancer-driven gene expression. Additionally, our study reveals a broad potential of CHD4 inhibition for cancer therapy and highlights chromatin remodelers as promising drug targets for cancer treatment.

## Results

### FP-RMS is particularly dependent on CHD4, amongst all NuRD subunits, for survival

Previously, we observed that CHD4 silencing leads to FP-RMS cell death *in vitro* and tumor regression *in vivo* (*Böhm et al., 2016*). Therefore, we investigated if other NuRD subunits are also required for the maintenance of FP-RMS cell viability. To this end, we established a NuRD-centered CRISPR/Cas9-based screen using the FP-RMS cell line RH4 in which we probed the most commonly described NuRD subunits (*Figure 1—figure supplement 1A*), including LSD1. We used five sgRNAs/gene and tested a total of 70 sgRNAs individually. CHD5 was excluded from this screen due to its preferential expression in neural and testicular tissues (*Kolla et al., 2015*). Indeed, RNA-seq data of RH4 cells demonstrated that CHD5 is not expressed (*Figure 1—figure supplement 1B*). Apart from MTA2, all other NuRD subunits tested are highly expressed in FP-RMS tumor tissue (*Figure 1—figure supplement 1C*). In brief, RH4 cells stably expressing Cas9 (*Figure 1—figure supplement 1D*) were transduced either with RFP-labelled sgRNAs targeting a given NuRD subunit, or a BFP-labelled control guide (sgAAVS1). Two days after transduction, the RFP and BFP populations were mixed 1:1, allowed to proliferate and their propagation was assessed on day 12 by flow cytometry (*Figure 1A*). The results of this screen are depicted in *Figure 1B* as ratio between knockout and control populations (RFP and BFP populations, respectively) normalized to day 2. For CHD4, HDAC1, and RBBP4 knockouts, 4 out of 5 sgRNAs and for GATAD2A 3 out of 5 sgRNAs decreased

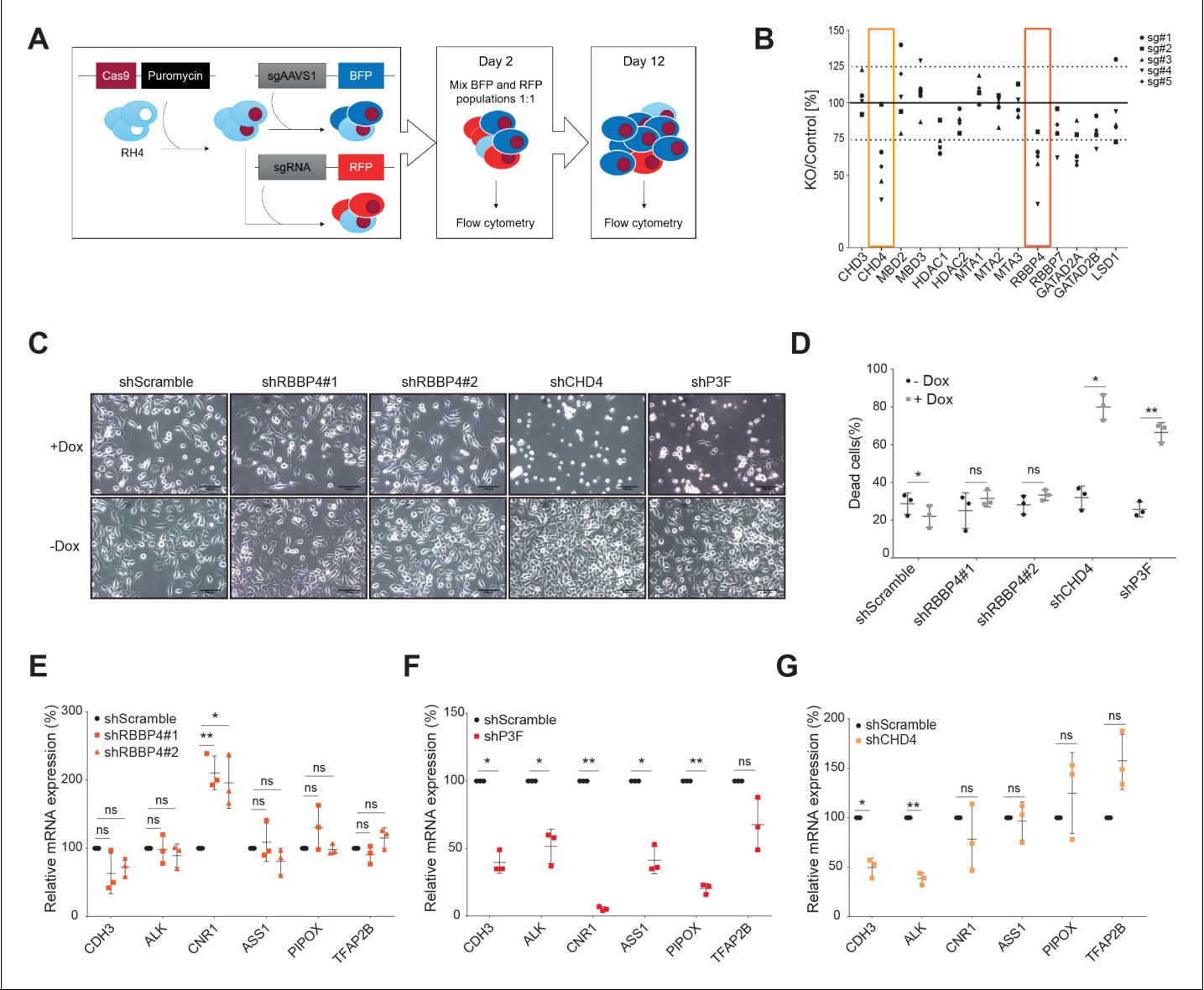

**Figure 1.** CHD4, unlike RBBP4, causes FP-RMS tumor cell death. (**A**) Illustrative scheme of the NuRD centered CRISPR/Cas9-based screen. Briefly, RH4 cells stably expressing Cas9 were transduced with lentiviral expression vectors containing either a BFP-labelled control sgRNA (sgAAVS1) or a RFP-labelled sgRNA targeting a certain NuRD subunit. Two days after transduction, the blue and red populations were mixed 1:1 and their evolution was analyzed by flow cytometry at day 2 and 12 after transduction. (**B**) CRISPR/Cas9 screen results displayed as ratio between the indicated NuRD member knockout (KO) population and the control population (RFP/BFB ratio) at day 12 normalized to day 2. Each point represents the average of 3 biological replicates. Five sgRNAs were used per NuRD member. (**C**) Representative phase-contrast images of RH4 cells 5 days after doxycycline-mediated (Dox) RBBP4, CHD4, and PAX3-FOXO1 (P3F) depletion by shRNA. A scramble shRNA was used as negative control. Scale bar - 100μm. (**D**) Percentage of dead cells, measured by 7-AAD staining, observed in the same samples described in (**C**). Data are represented as mean ± SD (n=3; *p< 0.1, **p < 0.01, ***p < 0.001, ratio paired t test). (**E, F and G**) Expression levels (relative to GAPDH) of the indicated P3F target genes quantified by qPCR in RH4 cells at 48hrs upon RBBP4, P3F and CHD4 induced knockdown by doxycycline treatment. Data were normalized to untreated cells and are represented as mean ± SD (n=3; *p<0.1, **p < 0.01, ***p < 0.001, ratio paired t test).

The online version of this article includes the following source data and figure supplement(s) for figure 1:

**Source data 1.** Raw data and statistics related to *Figure 1* and its supplements.
**Figure supplement 1.** The NuRD complex expression and function in FP-RMS.
**Figure supplement 2.** RBBP4 silencing reduces FP-RMS cell proliferation.

RH4 cell proliferation below the 75% threshold. Notably, knockouts of the MBD and MTA proteins, both core and mutually exclusive members of NuRD (*Allen et al., 2013*), as well as of LSD1 did not significantly alter RH4 cell proliferation at day 12. Analysis of a publicly available CRISPR-based genome-wide cancer vulnerability screen (CRISPR Avana Public 19Q2, depmap.org) for sensitivities to the depletion of NuRD subunits in 6 FP-RMS cell lines (RH28, RHJT, RH4, CW9019, JR, and RH30) confirmed the marked dependency of FP-RMS cells to CHD4 and RBBP4 (*Figure 1—figure supplement 1E*). Since the paralogs of the NuRD subunits might act redundantly, we performed double knockouts (DKO) of the MBD2/3, HDAC1/2, and GATAD2A/B paralogs (see Materials and methods) and investigated their effect in our competitive proliferation screen. All the combinations of guides tested reduced cell proliferation below the 75% threshold (median of DKO/Control ratio obtained with the five sgRNAs combinations tested: MBD2/3–68%, HDAC1/2–59%, GATAD2A/B – 65%; *Figure 1—figure supplement 1F*), suggesting a dependency of FP-RMS to NuRD. However, depletion of NuRD members as single or double knockouts affected FP-RMS cell proliferation to a lesser extent than CHD4 single knockout (median of KO/Control ratio obtained with the five sgRNAs tested: CHD3–98%, CHD4–51%, MBD2–102%, MBD3–107%, MBD2/3–68%, HDAC1–71%, HDAC2–89%, HDAC1/2–59%, MTA1–109%, MTA2–98%, MTA3–92%, RBBP4–66%, RBBP7–82%, GATAD2A – 63%, GATAD2B – 80%, GATAD2A/B – 65%, LSD1–88%; *Figure 1—source data 1*).

Next, we validated RBBP4 relevance for FP-RMS cell proliferation by establishing RH4 cell lines stably expressing two doxycycline-inducible shRNAs targeting RBBP4. Silencing of RBBP4 was confirmed on protein and mRNA levels after doxycycline treatment (*Figure 1—figure supplement 2A and B*). To compare RBBP4 with CHD4 and the fusion protein PAX3-FOXO1 (P3F) itself, we used similar and already established cells (*Böhm et al., 2016*) expressing doxycycline-inducible shRNAs targeting either CHD4 or P3F (knockdown validation is shown in *Figure 1—figure supplement 2C and D*). Confirming the CRISPR screen, RBBP4 depletion by shRNA reduced FP-RMS cell proliferation (*Figure 1—figure supplement 2E and F*). However, unlike CHD4 and P3F silencing, RBBP4 knockdown did not induce FP-RMS cell death (*Figure 1C and D*) nor did it influence the expression of 5 out of 6 selected P3F target genes (*Figure 1E,F and G*).

Taken together, our observations suggest that FP-RMS is particularly sensitive to CHD4 depletion amongst all NuRD subunits and that the reduced proliferation observed after RBBP4 loss occurs independently of suppression of the P3F signature.

## CHD4 interacts with negative and positive regulators of gene expression including BRD4

CHD4 does not recognize a specific DNA sequence (*Bouazoune et al., 2002*). Instead, it is recruited to the genome by its interaction partners. Hence, we decided to first define the interactome of this remodeler to better understand its activity in FP-RMS. To this aim, we introduced a 3xFlag tag in-frame at the N- and C-terminus of the endogenous *CHD4* gene via CRISPR/Cas9 mediated homologous repair in RH4 cells (*Figure 2—figure supplement 1A–D*) and performed affinity purification-mass spectrometry assays using an anti-Flag antibody to immunoprecipitated CHD4 (*Figure 2A*). Three independent Flag pull-downs were performed in both N- and C-terminus Flag-tagged RH4 cell lines with an average of 46% of bait coverage on unique peptide level. All experiments were carried out in the presence of benzonase to reduce the identification of DNA-mediated indirect interactions. Data from three control Flag immunoprecipitations in RH4 wildtype cells were also acquired and additional controls from CRAPome (*Mellacheruvu et al., 2013*) were added for the statistical scoring of interaction partners. Considering a fold change of at least two and a saint score higher than 0.6, a total of 103 potential interactors were identified (*Figure 2B*). To evaluate the quality of our interactome we compared it to reported interaction partners available on BioGRID 3.5 (*Oughtred et al., 2019*) and observed that 59% of our putative interactors have also been detected in previous publications (*Figure 2—source data 1*).

Gene ontology analysis of the putative interactors was performed using the Metascape online platform (http://metascape.org/) and revealed an enrichment for epigenetic regulators of gene expression (*Figure 2C*). After further categorization of the potential interactors, we confirmed that the majority are involved in transcription regulation and belong to chromatin remodeling or modifying complexes (*Figure 2D*). Importantly, since CHD4 strongly interacts with chromatin through its histone binding domains (*Mansfield et al., 2011*) and nucleosome core particles are present in our

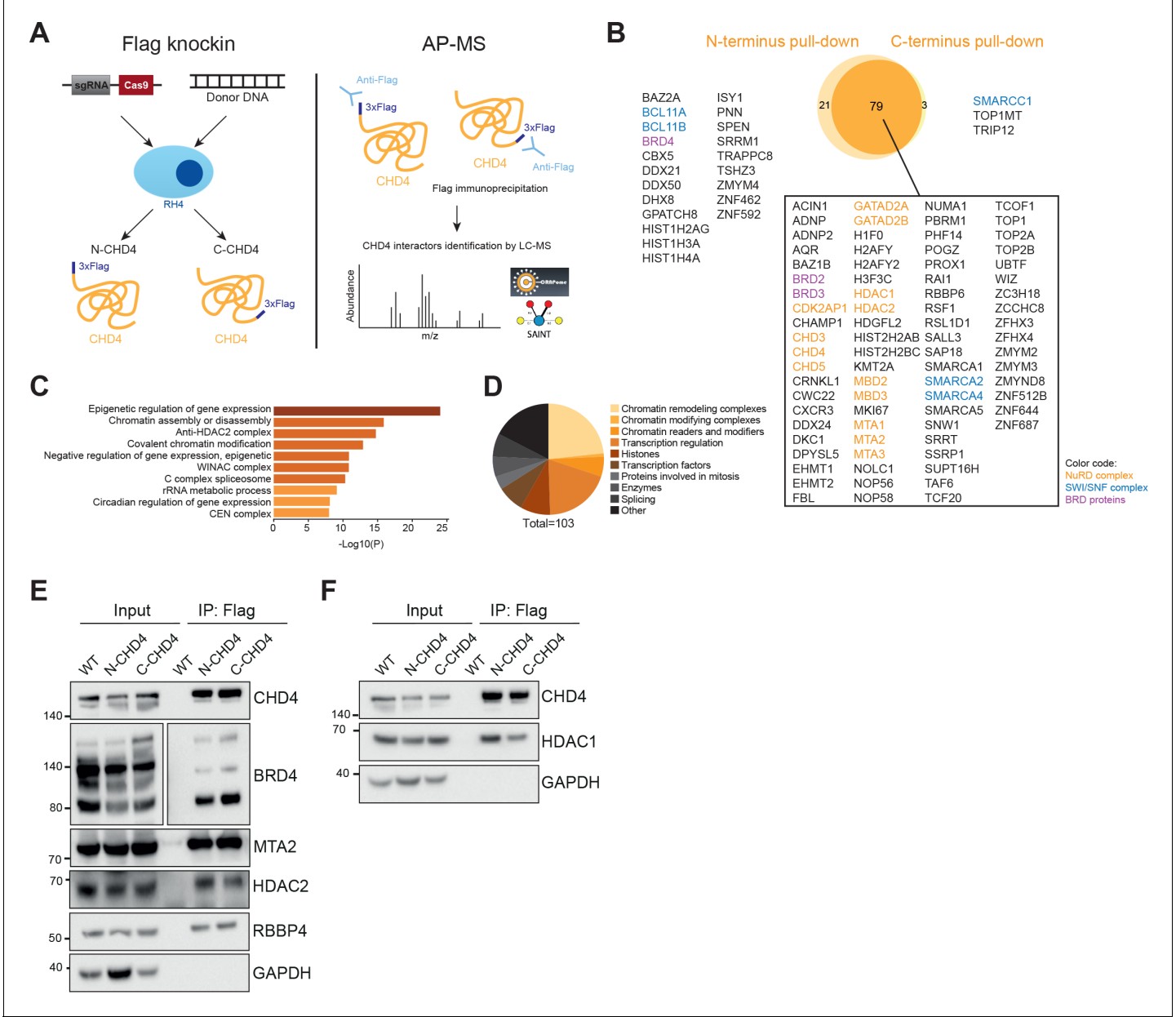

**Figure 2.** Mass spectrometry analysis of CHD4 interactome exposes interaction with the gene expression activator BRD4. (**A**) Illustrative scheme of the affinity purification-mass spectrometry (AP-MS) studies performed to identify CHD4 interactors. CRISPR/Cas9-mediated repair was used to endogenously Flag tag CHD4 on RH4 cells at the N- and C-terminus, creating two new clonal cell lines (N-CHD4 and C-CHD4) (left). Endogenous CHD4 was immunoprecipitated from the N- and C-CHD4 cell lines using an anti-Flag antibody and interactors were identified by liquid chromatography-mass spectrometry (LC-MS)(right). (**B**) Overlap of CHD4 putative interactors identified in the Flag pull-downs of CHD4. (**C**) Top 10 gene ontology terms found enriched on CHD4 interactome by Metascape online tool. (**D**) Distribution of the putative CHD4 interactors according to their protein class. (**E and F**) Western blots of Flag immunoprecipitation assays (IP).

The online version of this article includes the following source data and figure supplement(s) for figure 2:

Source data 1. List of CHD4 candidate interactors.

Figure supplement 1. CRISPR/Cas9-mediated repair efficiently inserts a 3xFlag tag to endogenous *CHD4* and *BRD4*.

interactome, we cannot exclude that some of the interactions identified here are indirect and chromatin-mediated.

Expectedly, the NuRD subunits, excluding RBBP4/7, were identified as high confidence interactors (*Figure 2B* and *Figure 2—source data 1*). The CHD4 paralogs, CHD3 and CHD5, were also

present in the interactome due to their homology to CHD4 (unique peptides identified: CHD3-10, CHD4-96, CHD5-2; shared peptides: 8). Also present in the list of putative interactors were several members of the SWI/SNF complex, the bromodomain-containing proteins BRD2/3/4 and the methyl-transferases EHMT1/2 and KMT2A (*Figure 2B*).

Since BRD4 is essential for the regulation of the aberrant P3F gene expression signature in FP-RMS (*Gryder et al., 2017*), we performed co-immunoprecipitation assays to confirm the CHD4-BRD4 interaction. Flag immunoprecipitation of CHD4 was able to pull-down BRD4 as well as all sub-units of NuRD tested, including RBBP4, which was not identified in our interactome studies (*Figure 2E and F*). For the reverse immunoprecipitation, we endogenously inserted a 3xFlag tag at the N-terminus of *BRD4* in RH4 cells (*Figure 2—figure supplement 1B,C and E*). Interestingly, the Flag pull-down of BRD4 not only coprecipitated CHD4 but also HDAC2 and MTA2 (*Figure 2—figure supplement 1F and G*), suggesting that BRD4 interacts with CHD4 in the context of NuRD. How-ever, BRD4 immunoprecipitation was not able to pull-down HDAC1 nor RBBP4 (*Figure 2—figure supplement 1G and H*), which can potentially reflect the composition of the NuRD complex that integrates BRD4 or can be suggestive of the BRD4 position within the NuRD complex.

In conclusion, the analysis of CHD4 interactome indicates that this remodeler has a complex func-tion in FP-RMS. On one hand, it interacts with many transcription activators such as members of the SWI/SNF complex and the acetylation reader BRD4, but on the other, high confidence interactions with known transcription repressors, like EHMT1/2, were also detected. These diversity of interaction partners might potentially modulate the activity of CHD4 as a transcription activator or repressor in a context-dependent manner.

## CHD4/NuRD localizes to enhancers while CHD4-free NuRD to promoters

To investigate the function of the NuRD complex on the genome level, we performed ChIP-seq assays in RH4 cells for CHD4, RBBP4, HDAC2, and MTA2, as well as for the tumor driver P3F, and other relevant epigenetic regulators and histone marks. Additionally, DNase I hypersensitivity assays (DNase) were completed to evaluate genome accessibility (see Materials and methods for accession numbers).

A correlation matrix generated from the ChIP-seq data (*Figure 3A*) demonstrated that CHD4 co-occurs with its interactors RBBP4, HDAC2, MTA2, and BRD4 in the genome. Strikingly, CHD4 ChIP-seq signal also correlated with the one of P3F and the enhancer marks H3K27ac and H3K4me1, but not with H3K27me3. The overlay of the ChIP-seq signals of CHD4, RBBP4, HDAC2, and MTA2 on the chromatin states map (*Ernst et al., 2011*) confirmed that these NuRD members mainly localized to enhancers (*Figure 3B*). Interestingly, RBBP4, HDAC2, and MTA2, in contrast to CHD4, showed an additional strong prevalence at active promoters.

The overlap of the genomic locations of the NuRD components (*Figure 3C* and *Figure 3—source data 1*) showed that roughly 66% of the locations shared amongst HDAC2, RBBP4 and MTA2 (8901 out of 13,5000) did not co-localize with CHD4, suggesting the presence of a CHD4-free NuRD com-plex (NuRD-only) and a NuRD complex containing CHD4 (CHD4/NuRD). The NuRD-only peaks (n = 8,901) were frequently found in the vicinity of transcription start sites (TSSs) and in promoter regions (*Figure 3D*), in contrast to CHD4/NuRD peaks (n = 4,599) which predominantly located dis-tally to TSSs and to intronic or intergenic regions (*Figure 3—source data 1*). In line with these results, NuRD-only locations were characterized by the presence of RNA Pol 2 and the promoter mark H3K4me3, while CHD4/NuRD sites were richer in the enhancer-related histone marks H3K4me1 and H3K27ac, as well as in BRD4 (*Figure 3E and F*). As expected for active regions, both NuRD-only and CHD4/NuRD locations were sensitive to DNase I digestion, suggesting an open chromatin conformation at these sites (*Figure 3E and F*). Examples of a CHD4/NuRD enhancer and a NuRD-only promoter tracks are depicted in *Figure 3G*.

The presence of CHD4/NuRD at these 4,599 locations distal to TSSs identified in RH4 cells was confirmed in two other FP-RMS cell lines by performing ChIP-seq assays for CHD4, RBBP4, HDAC2, and MTA2 in RH5 and SCMC cells (35% and 54% of the 4,599 CHD4/NuRD locations identified in RH4 were also occupied by CHD4/NuRD in RH5 and SCMC cells, respectively; *Figure 3—figure sup-plement 1A*). As observed in RH4 cells, in SCMC cells NuRD without CHD4 was also more com-monly found closer to TSSs and at promoter regions than CHD4-free NuRD, although this difference was minimal in RH5 cells (*Figure 3—figure supplement 1B*).

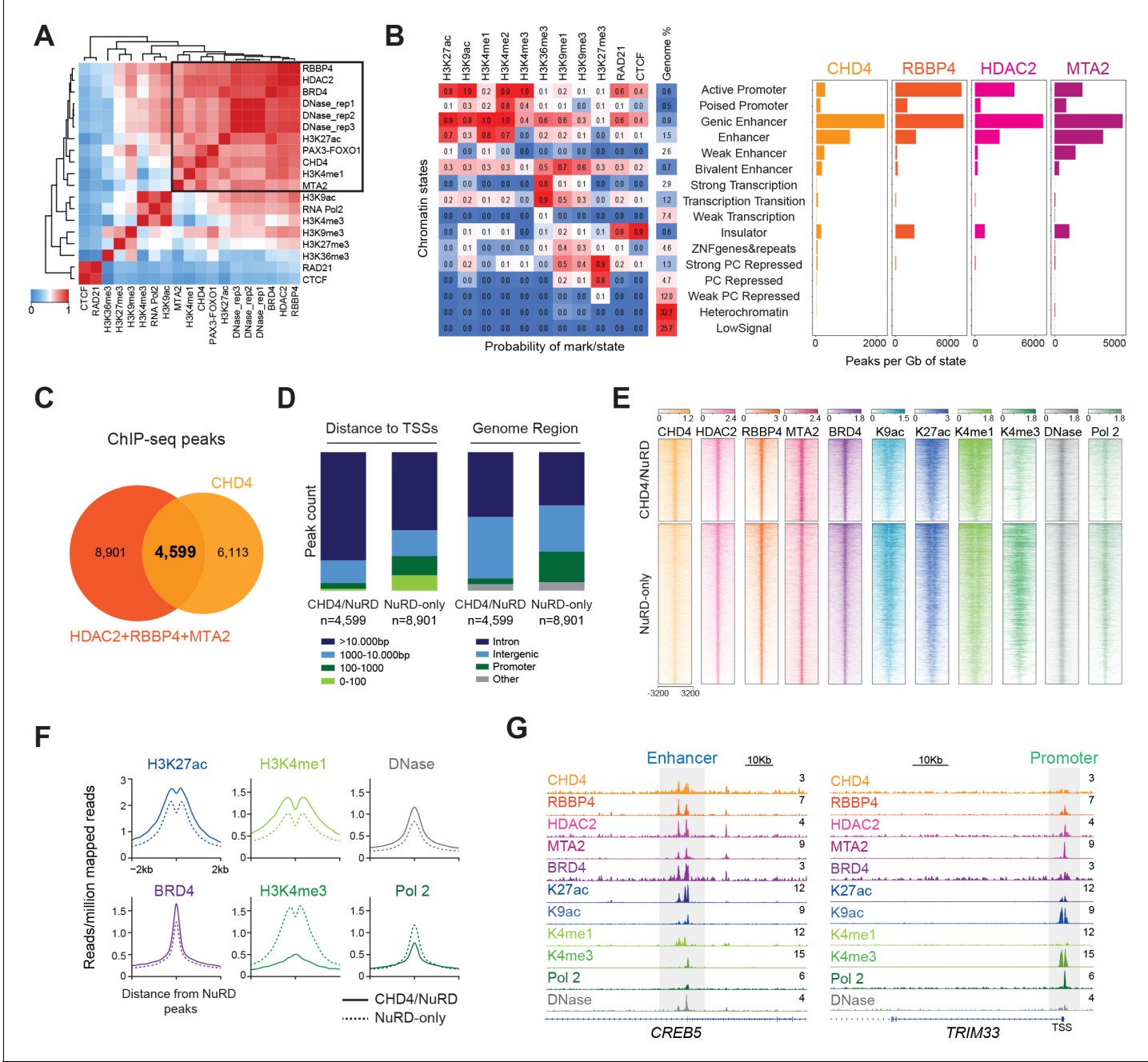

**Figure 3.** NuRD localizes to active chromatin with distinct compositions at enhancers and promoters. (**A**) Pearson correlation heatmap of DNase I hypersensitivity (DNase) and ChIP-seq signal of the indicated epigenetic factors and histone marks in RH4 cells. Datasets are ordered by unsupervised clustering. (**B**) Chromatin states and respective abundance of the depicted NuRD components per state. (**C**) Overlap of CHD4, RBBP4, MTA2, and HDAC2 ChIP-seq peaks. (**D**) Distribution of the peak counts for CHD4/NuRD and NuRD-only regions according to their distance to the transcription start sites (TSSs) and genome functional region. (**E**) Heatmap depicting the ChIP-seq signal of the indicated NuRD subunits, BRD4, histone marks (H3K9ac,H3K27ac, H3K4me1, and H3K4me3), RNA Polymerase 2 (Pol 2), and DNase I hypersensitivity signal at CHD4/NuRD (n=4,599) and NuRD-only regions (n=8,901). The rows show 8kb regions, centered on HDAC2 peaks and ranked by the ChIP-seq signal intensity of H3K27ac. Color shading corresponds to ChIP-seq read counts. (**F**) Density plots displaying the average ChIP-seq signal of H3K27ac, H3K4me1, BRD4, H3K4me3, RNA Polymerase 2, and DNase I hypersensitivity signal at CHD4/NuRD and NuRD-only locations. (**G**) Examples of gene tracks displaying the ChIP-seq signal of the indicated proteins, histone marks and DNase I hypersensitivity signal at a CHD4/NuRD enhancer (*CREB5*) and a NuRD-only promoter (*TRIM33*). The online version of this article includes the following source data and figure supplement(s) for figure 3:

**Source data 1.** NuRD ChIP-seq locations.

**Figure supplement 1.** The NuRD complex localizes to similar enhancer locations in RH5 and SCMC cells as in RH4.

*Figure 3 continued on next page*

*Figure 3 continued*

**Figure supplement 2.** The NuRD complex regulates distinct processes according to the presence or absence of CHD4.

In summary, these ChIP-seq assays demonstrate that the NuRD complex in FP-RMS is present at active genomic regions but with distinct compositions at enhancers and promoters. At enhancers, the NuRD complex integrates the chromatin remodeler CHD4 while at promoters CHD4 was normally absent.

## NuRD regulates distinct biological processes according to its composition and location

To functionally distinguish NuRD-only and CHD4/NuRD locations, we performed separate gene ontology analysis for these regions using the online tool GREAT (*McLean et al., 2010*). The top 15 biological processes found enriched in these two sets of genomic locations are shown in *Figure 3— figure supplement 2A*. After categorization of all biological processes found, we observed that CHD4/NuRD regions were associated with processes involved in the regulation of muscle development and other myogenic processes more frequently than NuRD-only regions (*Figure 3—figure supplement 2A*). An example of a CHD4/NuRD location associated with the expression of a myogenic specific gene (*MYLK2* - myosin light chain kinase 2) is shown in *Figure 3—figure supplement 2B*. NuRD-only locations were instead correlated with the regulation of genes involved in general processes such as gene expression and mRNA metabolism (*Figure 3—figure supplement 2A*).

These findings propose that NuRD regulates distinct biological processes according to its location and the presence of CHD4. NuRD-only peaks, including many promoter regions, were associated with the regulation of housekeeping processes, whereas CHD4/NuRD locations, predominantly found in enhancers, were involved in the regulation of cell-type specific processes characteristic of FP-RMS.

## CHD4 binds to P3F-containing super-enhancers and allows an open chromatin conformation at these regulatory regions

Super-enhancers (SEs) are enhancer clusters abundantly populated by transcription factors and cofactors. During normal development, SEs regulate cell identity while in cancer they drive high expression of oncogenes (*Hnisz et al., 2013*; *Sengupta and George, 2017*). In FP-RMS, P3F drives tumorigenesis by creating and binding to SEs to alter gene expression (*Gryder et al., 2017*). Since we observed that CHD4/NuRD enhancer locations potentially regulate the expression of cell-type specific genes, we investigated the involvement of these regions in P3F- and SE-mediated oncogenesis. The overlap between P3F and CHD4/NuRD ChIP-seq signal (*Figure 4A* and *Figure 4—source data 1*) showed that 42% of P3F locations were also bound by CHD4/NuRD (P3F+CHD4/NuRD regions; 1,538 out of 3,696 peaks). Interestingly, P3F+CHD4/NuRD sites, in comparison with P3F-only locations (n = 2,158), were richer in the enhancer mark H3K27ac and BRD4, and displayed a more open conformation as shown by DNase I hypersensitivity assays (*Figure 4B*). GREAT ontology analysis of the P3F-only, P3F+CHD4/NuRD and P3F-free CHD4/NuRD regions (CHD4/NuRD-only regions) demonstrated that cell type-specific processes related with muscle development, which are characteristic of FP-RMS, were mainly associated with P3F+CHD4/NuRD and P3F-only regions and absent from the top 15 enriched biological processes obtained for CHD4/NuRD-only regions (*Figure 4—figure supplement 1*). Analysis of the expression of the nearest genes, within topological associated domains (TADs), associated with P3F-only, P3F+CHD4/NuRD and CHD4/NuRD-only binding sites (*Figure 4—source data 1*) revealed that P3F+CHD4/NuRD-regulated genes were significantly higher expressed than P3F-only-regulated genes (*Figure 4C*). The co-localization of P3F and CHD4/NuRD in FP-RMS was further confirmed by performing P3F ChIP-seq assays (using breakpoint-specific antibody, *Cao et al., 2010*) in RH5 and SCMC cells. P3F and CHD4/NuRD locations were defined by the presence of ChIP-seq signal in at least two out of the three FP-RMS cell lines tested (RH4, RH5 and SCMC). Using this criterion, we observed that CHD4/NuRD co-localized with P3F in roughly 50% of P3F binding sites (778 out of 1,569; *Figure 4—figure supplement 2A and B*).

Regarding the presence of CHD4/NuRD at SEs, we observed that HDAC2, RBBP4, and MTA2 were present in a total of 784 out of the 810 SEs predicted according to H3K27ac abundancy in RH4

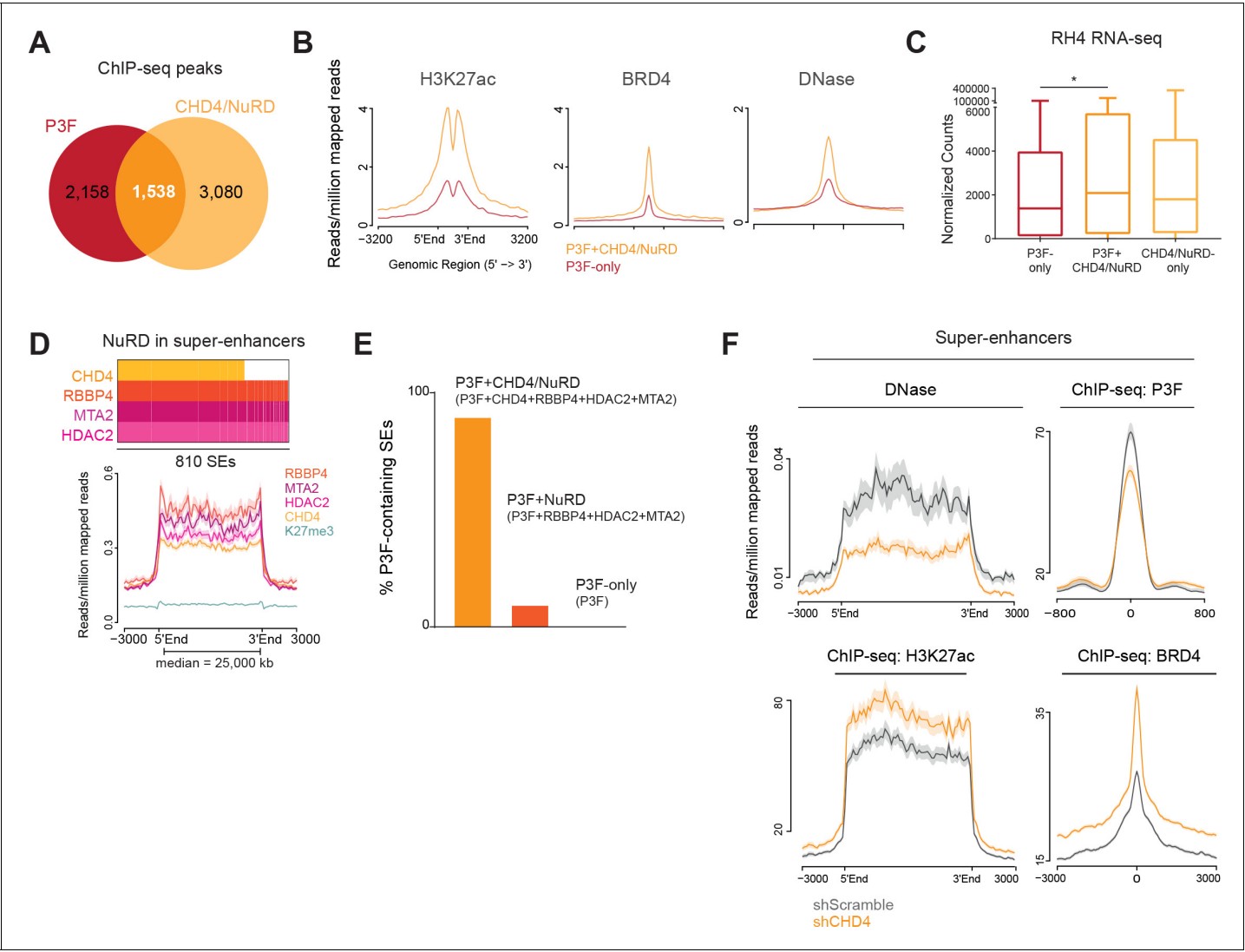

**Figure 4.** CHD4 influences chromatin accessibility and allows P3F binding to SEs. (**A**) Overlap between P3F and CHD4/NuRD ChIP-seq peaks. (**B**) Density plots depicting the average H3K27ac and BRD4 ChIP-seq as well as DNase I hypersensitivity (DNase) signal in RH4 cells at P3F+CHD4/NuRD (orange, n=1,538) and P3F-only locations (red, n=2,158). (**C**) Expression levels as normalized counts, obtained from RNA-seq data of RH4 cells, of the genes located nearest, within TADs, to P3F-only, P3F+CHD4/NuRD and CHD4/NuRD-only locations (one-way ANOVA; adjusted p-value=0.0411; *p< 0.1, **p < 0.01, ***p < 0.001). (**D**) Representative plot of the presence of the indicated NuRD subunits at the 810 super-enhancers (SEs) identified in RH4 cells (top). Density plot showing the average ChIP-seq signal of RBBP4, MTA2, HDAC2, CHD4, and H3K27me3 at SEs (bottom). (**E**) Distribution (in percentage) of P3F-bound SEs according to the presence of NuRD subunits. (**F**) Density plots depicting the average DNase I hypersensitivity signal, P3F, H3K27ac and BRD4 ChIP-seq signal in RH4 cells at SEs upon 48hrs of CHD4 knockdown (orange).

The online version of this article includes the following source data and figure supplement(s) for figure 4:

**Source data 1.** PAX3-FOXO1 and CHD4/NuRD co-occupancy at enhancers and SEs.
**Figure supplement 1.** PAX3-FOXO1 regulates muscle-related processes with CHD4/NuRD.
**Figure supplement 2.** CHD4/NuRD is present at SEs and co-localizes with a subset of P3F locations in RH5 and SCMC cells.
**Figure supplement 3.** CHD4 depletion impairs P3F binding to enhancers and SEs.

cells (*Figure 4D* and *Figure 4—source data 1*). Of these 784 SEs, 595 were co-bound by CHD4 (*Figure 4D* and *Figure 4—source data 1*). We confirmed the presence of CHD4/NuRD at these 810 SEs in both RH5 and SCMC cells (*Figure 4—figure supplement 2C*). Remarkably, analysis of the SEs occupied by P3F in RH4 cells demonstrated that 90% (407 out of 452) were co-bound by CHD4/ NuRD (*Figure 4E* and *Figure 4—source data 1*), suggesting that CHD4/NuRD plays a relevant role in the regulation of SEs bound by P3F.

Since CHD4 is a nucleosome remodeler capable of moving nucleosomes along the DNA (*Xue et al., 1998*), we hypothesized that CHD4 might influence the chromatin architecture at its binding sites. To investigate this in detail, we performed DNase I hypersensitivity assays upon 48hrs of CHD4 depletion in RH4 cells. Strikingly, we observed a drastic decrease in SE accessibility upon CHD4 silencing (*Figure 4F*), suggesting that this chromatin remodeler is necessary for the maintenance of DNA accessibility at these locations. Similarly, at P3F+CHD4/NuRD locations (including enhancers and SEs) and CHD4/NuRD-only locations, CHD4 silencing led to a moderate decrease in genome accessibility (*Figure 4—figure supplement 3A*). Next, we assessed whether these alterations in chromatin architecture caused by CHD4 depletion would change the genomic localization of P3F and NuRD. To this end, we performed ChIP-seq assays with a P3F breakpoint-specific antibody (*Cao et al., 2010*) upon 48hrs of CHD4 silencing. Using spike-in normalization, we observed that CHD4 knockdown reduced P3F binding to SEs (*Figure 4F*). A comparable reduction in P3F binding was also observed at P3F+CHD4/NuRD locations (*Figure 4—figure supplement 3B*). To evaluate NuRD chromatin positioning upon CHD4 silencing, we performed ChIP-qPCR assays of HDAC2 at three P3F selected target locations and observed that HDAC2 binding was reduced at 2 of these locations (*ALK* and *CDH3*, *Figure 4—figure supplement 3C*). Curiously, both *ALK* and *CDH3* expression are reduced after CHD4 silencing but not the one of *ASS1* (*Figure 1G*). In line with HDAC2 displacement, we also observed that CHD4 depletion caused an increase and spread of H3K27ac as well as an increase of BRD4 binding to SEs (*Figure 4F*). Gene tracks illustrating the presence of CHD4/NuRD at P3F-bound SEs as well as changes in SE accessibility and P3F positioning upon CHD4 silencing are displayed in *Figure 4—figure supplement 3D and E*.

These results show that CHD4/NuRD occupies the more accessible and active P3F locations. In fact, nearly all P3F-containing SEs are co-bound by CHD4/NuRD. Additionally, we demonstrate for the first time that CHD4 regulates chromatin architecture at SEs and is essential to keep these *cis*-regulatory regions open and permissive to the binding of the oncogenic driver P3F.

## CHD4 regulates SE-mediated gene expression and influences RNA Pol 2 promoter binding

To evaluate the impact of CHD4 on P3F- and SE-mediated gene expression, we performed RNA-seq experiments 24hrs and 48hrs after doxycycline-induced silencing of CHD4 and P3F. We observed that silencing of either protein led to an equal number of up- and downregulated genes (after 48hrs of CHD4 silencing 2,195 genes were upregulated and 2,848 downregulated, while upon 48hrs of P3F silencing 1,827 were upregulated and 1,700 downregulated; false discovery rate - 1%, fold change ≥25%, *Figure 5A* and *Figure 5—figure supplement 1A*). Interestingly, both CHD4 and P3F depletion preferentially affected SE-mediated gene expression (*Figure 5B*). At 48hrs, CHD4 silencing influenced the expression of 28% of P3F target genes (541 protein coding genes were co-upregulated upon CHD4 or P3F depletion and 449 were co-downregulated, *Figure 5C and D*, *Figure 5—source data 1*). GSEA ontology analysis performed with the 990 CHD4 and P3F co-regulated genes was able to identify described P3F signatures and confirmed that P3F-enhancer regulated target genes were downregulated upon CHD4 silencing (*Figure 5E*). Importantly, we observed that CHD4 does not regulate the expression of P3F itself (*Figure 1—figure supplement 2C and D*), hence results obtained after CHD4 knockdown are not indirectly caused by reduction of the fusion protein levels.

Next, we sought to investigate if CHD4 also influenced RNA Pol 2 positioning. To this end, we examined total RNA Pol 2 binding in the presence and absence of CHD4 by ChIP-seq, using spike-in normalization, and observed that depletion of CHD4 decreased RNA Pol 2 binding at TSSs and transcription end sites (TESs) of genes co-downregulated by CHD4 and P3F suppression (n = 449, *Figure 5F*). This effect was specific to the downregulated genes since similar analysis performed with the co-upregulated genes (n = 541) resulted in an increase in RNA Pol 2 binding at TES (*Figure 5—figure supplement 1B*). Gene tracks illustrating changes in RNA Pol 2 binding upon CHD4 depletion are depicted in *Figure 5—figure supplement 1C and D*.

Together, our observations imply that CHD4 regulates SE-mediated gene expression and that it collaborates with P3F to activate gene expression partially by increasing the binding of RNA Pol 2 to the TSSs.

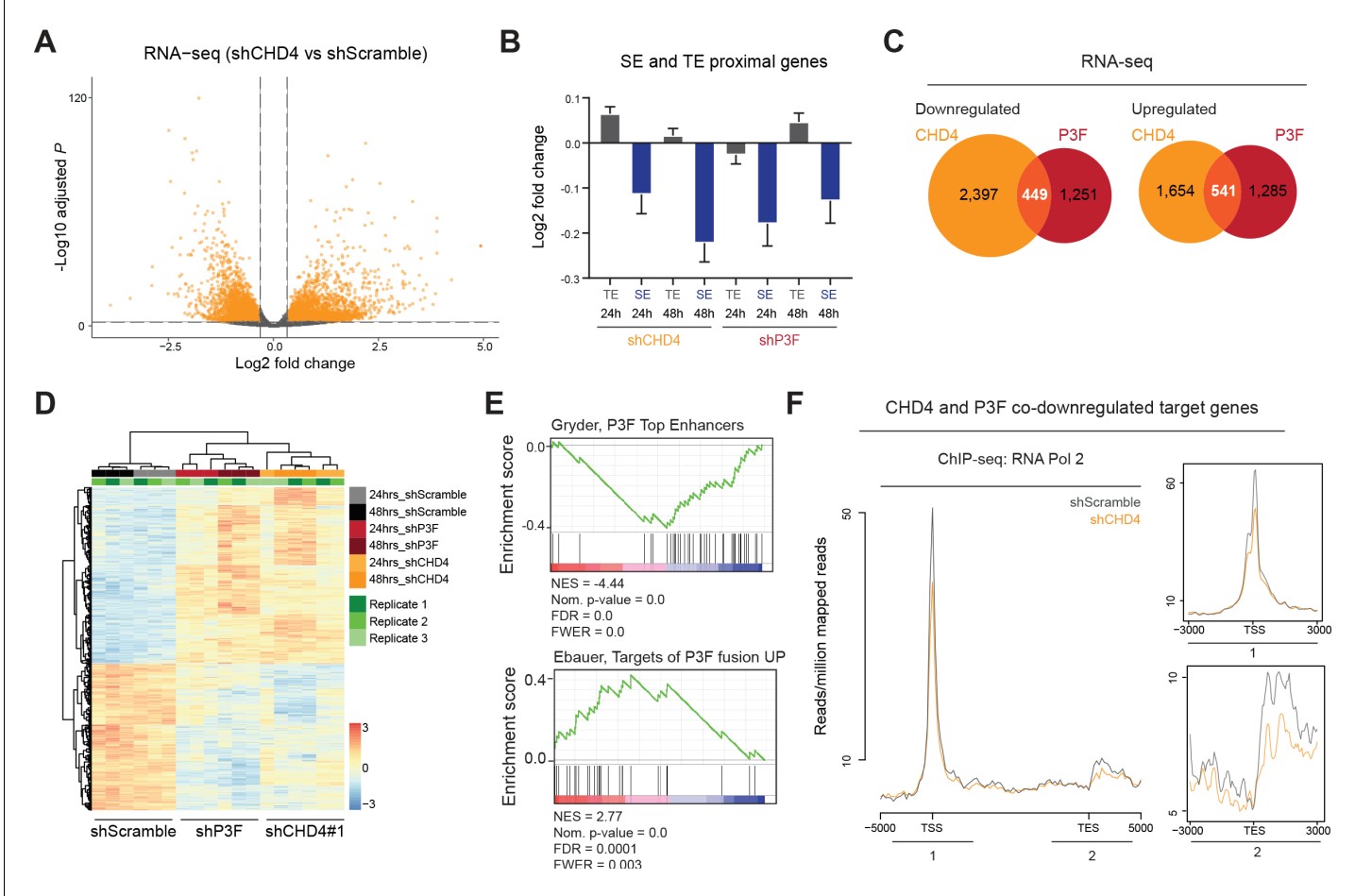

**Figure 5.** CHD4 regulates P3F- and SE-mediated gene expression as well as RNA Pol 2 binding to promoters. (A) Volcano plot depicting changes in gene expression upon 48hrs of CHD4 silencing in RH4 cells (fold change≥ 25%, false discovery rate of 1%). (B) Changes in expression, as log2 fold change, of the nearest genes within TADs associated with typical enhancers and super-enhancers (TE and SE, respectively) in RH4 cells upon 24 or 48hrs of CHD4 or P3F silencing. Data are represented as mean ± SEM. (C) Overlap of CHD4 and P3F regulated genes identified by RNA-seq upon 48hrs of silencing. (D) Heatmap of unsupervised hierarchical clustering analysis depicting CHD4 and P3F co-regulated genes (n=990) in RH4 cells. (E) GSEA ontology analysis performed with the CHD4 and P3F co-regulated signature (n=990) as pre-ranked dataset. NES – normalized enrichment score, FDR – false discovery rate, FWER – family-wise error rate. (F) Density plots depicting the average RNA Pol 2 ChIP-seq signal upon 48hrs of CHD4 silencing (orange) at genes co-downregulated by P3F and CHD4 (n=449).

The online version of this article includes the following source data and figure supplement(s) for figure 5:

**Source data 1.** CHD4 and PAX3-FOXO1 co-regulated target genes.

**Figure supplement 1.** RNA Pol 2 positioning is affected by CHD4 silencing.

## Large-scale genome screens suggest CHD4 as a broad tumor dependency

Besides FP-RMS, CHD4 has been implicated in the viability of other tumors such as breast cancer (*D'Alesio et al., 2016*), acute myeloid leukemia (*Heshmati et al., 2016*), lung cancer (*Xu et al., 2016*), and colorectal cancer (*Xia et al., 2017*). In agreement, analysis of the R2 gene expression database (r2.aml.nl) revealed that CHD4 expression is generally higher in tumors (269 datasets) than in normal tissue (38 datasets; *Figure 6A and B*). Therefore, we questioned if CHD4 constitutes a general tumor dependency. To answer this, we analyzed data from two genome-wide cancer genetic vulnerability screens available on the depmap online platform (depmap.org): the combined RNAi (*Marcotte et al., 2016*; *McDonald et al., 2017*; *Tsherniak et al., 2017*) and the CRISPR (Avana) Public 19Q2 (*Figure 6C*). The combined RNAi dataset comprises data from 974 RNAi screens targeting 17,309 genes in 712 cancer cell lines while the CRISPR dataset screened 17,634 genes in 558

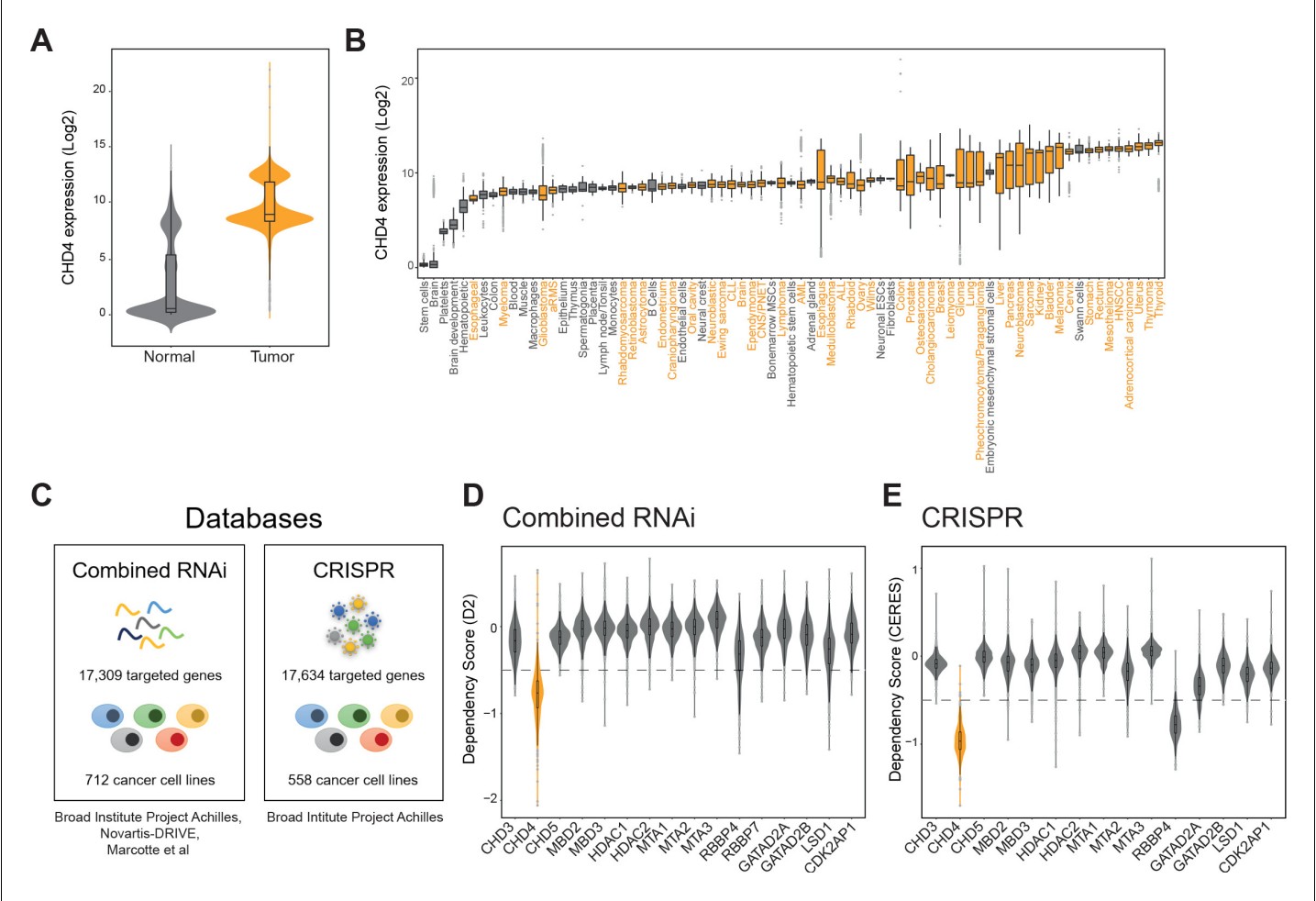

**Figure 6.** CHD4 is essential for a broad range of tumors. (**A and B**) Violin and boxplots depicting CHD4 expression levels (data: r2.aml.nl) in normal (grey) and tumor tissue (orange). (**C**) Databases used to evaluate tumor sensitivities to CHD4 silencing or knockout. (**D and E**) Violin plots showing the tumor dependency scores, calculated by D2 or CERES, of the indicated NuRD members. CHD4 is displayed in orange. The -0.5 threshold is depicted as a dashed line.

The online version of this article includes the following figure supplement(s) for figure 6:

**Figure supplement 1.** CHD4 depletion affects the viability of a variety of tumor types.

cancer cell lines. In both datasets, cancer vulnerability is depicted as a dependency score which estimates gene dependency on an absolute scale where zero represents no dependency or nonessentiality.

First, we analyzed all NuRD complex members present in both datasets (***Figure 6D and E***) using a threshold of -0.5 to determine cancer dependency. We observed that loss of CHD4 impaired tumor cell viability in >92% of the cancer cell lines used in both datasets (in the combined RNAi database, 627 out of 712 cancer cell lines were sensitive to CHD4 silencing and in the CRISPR dataset 552 out of 558 cancer cell lines were affected by CHD4 knockout). All tumor types represented in both datasets were sensitive to CHD4 depletion (***Figure 6—figure supplement 1A and B***). Other NuRD members, apart from RBBP4, showed only tumor-specific dependencies (***Figure 6D and E***). Next, we compared CHD4 with other SNF2-like chromatin remodelers (***Flaus et al., 2006***) and the broad cancer susceptibility gene *BRD4*. In both databases loss of SRCAP, EP400, INO80 and BRD4 induced a general impairment of tumor cell viability, however, the lowest dependency scores were observed for CHD4 (***Figure 6—figure supplement 1C and D***).

In summary, CHD4 is highly expressed in cancer and is essential not only for FP-RMS tumor cell survival but also for multiple tumor types. Therefore, our findings strongly suggest that CHD4 represents a promising new general target for cancer therapy, in agreement with its function as a regulator of SE activity.

## Discussion

Cancer-specific chromosomal aberrations producing chimeric fusion genes are recurrently found in pediatric sarcomas. In FP-RMS, the transcription factor PAX3-FOXO1 is the product of such fusion gene and it is commonly perceived as the founding genetic abnormality driving the development of this malignancy by changing gene expression. Since direct targeting of transcription factors is still very challenging, acting on the activity of PAX3-FOXO1 at the chromatin level presents a robust alternative for FP-RMS therapy. To this end, we studied here in detail the mechanisms by which the chromatin remodeler CHD4, in the context of the NuRD complex, influences P3F-regulated gene expression and FP-RMS cell viability.

First, we investigated the relevance of the most commonly described NuRD members individually for FP-RMS cell proliferation. Our CRISPR/Cas9 screen revealed that amongst the enzymatic members of NuRD (CHD3/4, HDAC1/2, and LSD1) only CHD4 knockout strongly impaired FP-RMS cell proliferation. The lack of response observed upon LSD1 knockout, at 12 days, is in line with our previous work where we demonstrated that LSD1 silencing by shRNA did not perturb the viability of FP-RMS cells (*Böhm et al., 2016*). The single knockouts as well as the double knockouts of the redundant NuRD subunits HDAC1/2 (*Jurkin et al., 2011*) resulted in a consistent although moderate decrease in FP-RMS cell proliferation, suggesting that both chromatin remodeling and histone deacetylation are essential functions of NuRD for FP-RMS viability. In fact, a specific HDAC1/2 inhibitor was shown to decrease FP-RMS proliferation and disrupt SE-driven gene expression (*Gryder et al., 2019b*) However, since HDAC1/2 are found in other complexes, such as SIN3 and CoREST (*Kelly and Cowley, 2013*), a NuRD-independent function of these enzymes cannot be excluded. Amongst the non-enzymatic members of NuRD, RBBP4 knockout led to a considerable reduction of FP-RMS cell proliferation. Yet, RBBP4 is also present in other chromatin modifying complexes, such as SIN3 and PRC2 (*Allen et al., 2013*), and its silencing led to a phenotype distinct from the one obtained upon CHD4 depletion. The knockouts of GATAD2A/B also caused consistent although modest decreases in FP-RMS cell proliferation. These paralogs connect CHD4 to the MBD-GATAD2 dimer which binds to the HDAC-MTA-RBBP subcomplex, forming the NuRD complex. Interestingly, immunoprecipitation of GATAD2A in mammalian cells pulls-down CHD4 but no other NuRD component (*Torrado et al., 2017*) and in mouse embryonic stem cells in the absence of Mbd3 Chd4 remains associated with Gatad2b (*Bornelöv et al., 2018*), suggesting a strong interaction between CHD4 and the GATAD2 paralogs and possibly a functional dependence of CHD4 on them. Despite single knockouts of the MBD subunits, which are mutually exclusive members of the complex and crucial for the assembly NuRD (*Zhang et al., 2016*), did not interfere with FP-RMS cell proliferation, MBD2/3 double knockout consistently reduced the proliferation of FP-RMS cells. These findings indicate that the assembly of NuRD is necessary for FP-RMS proliferation although only CHD4 depletion had such strong effect on cell viability as single knockout. Additionally, our interactome studies suggest that CHD4 might collaborate with other chromatin remodeling complexes besides NuRD, such as SWI/SNF, and transcription regulators, like BRD4, to influence FP-RMS cell proliferation.

CHD4 is a nucleosome remodeler able to change DNA accessibility and influence gene expression. Therefore, to study the mechanism behind FP-RMS dependency on CHD4, we performed ChIP-seq assays to localize CHD4 in the genome and DNase I hypersensitivity assays to further understand the contribution of CHD4's remodeling ability for the regulation of P3F-mediated gene expression. We observed that CHD4 binds to active enhancers related to the regulation of cell-type specific processes like muscle development together with P3F and the NuRD subunits RBBP4, HDAC2, and MTA2. In addition, the vast majority of P3F-bound SEs were co-bound by CHD4. Mechanistically, knockdown of CHD4 drastically reduced DNA accessibility at SEs which interfered with P3F binding to these *cis*-regulatory elements and resulted in a reduction of SE-regulated gene expression (see model *Figure 7*). CHD4 silencing also displaced RNA Pol 2 from promoters and removed HDAC2 from the chromatin which consequently led to an increase and spread of the acetylation marker

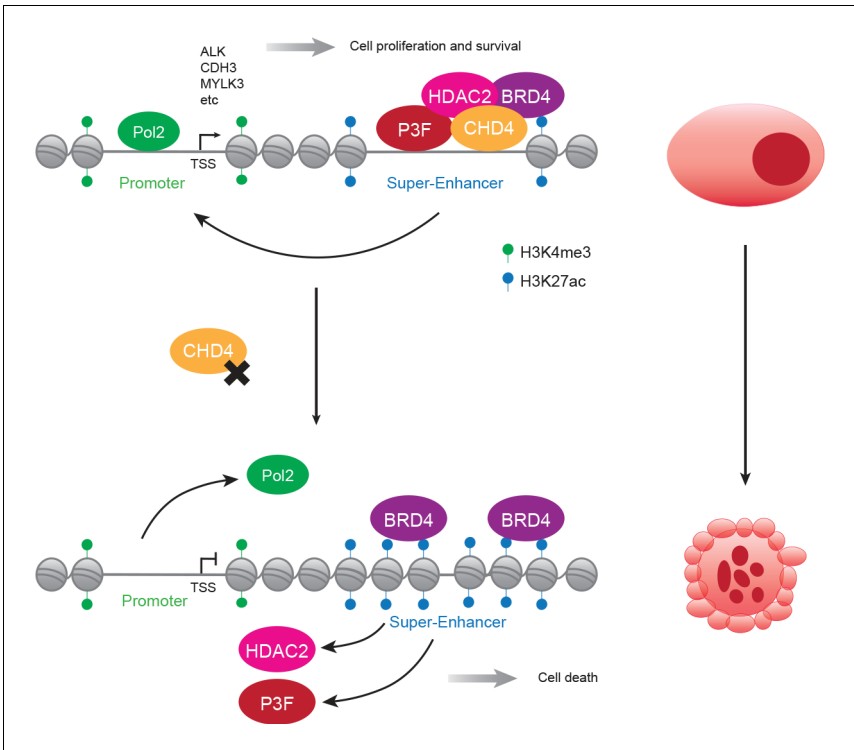

**Figure 7.** Proposed model of CHD4-dependent and P3F-driven gene expression regulation. In FP-RMS, CHD4/NuRD co-localizes with P3F and BRD4 at enhancers and super-enhancers enabling the expression of a subset of the fusion protein target genes and allowing tumor maintenance and survival (top). In the absence of CHD4, super-enhancers lose DNA accessibility and, consequentially, binding of P3F and HDAC2 which leads to a spread of H3K27ac, an increase in BRD4 binding and prevents the positioning of RNA Pol 2 to promoters (bottom). These changes of chromatin architecture result in a reduction of SE- and P3F-regulated gene expression contributing to tumor cell death.

H3K27ac at SEs resulting in an increase in BRD4 binding to these regions. Such correlation between reduced HDAC activity, acetylation spreading and RNA Pol 2 depletion halting transcription initiation has been recently described (*Gryder et al., 2019a*). These results confirm that both the remodeling activity and histone deacetylation are core functions of NuRD required for the expression of genes regulated by SEs in FP-RMS.

CHD4 silencing was also responsible for the upregulation of several genes and we detected high confidence interactions between this chromatin remodeler and transcription repressors. Hence, further studies investigating the role of CHD4 in repression of gene expression in the context of FP-RMS are necessary to fully understand the function of this remodeler in the expression signature of this tumor.

Besides this pediatric malignancy, CHD4 is essential for the survival of a broad range of tumor types (*Chudnovsky et al., 2014*; *D'Alesio et al., 2016*; *Heshmati et al., 2016*). Consistent with these reports, we found that CHD4 is highly expressed in cancer and analysis of two publicly available databases of tumor susceptibilities identified CHD4 as an essential gene in a variety of tumors, a characteristic that stood out among the other NuRD subunits and SNF2-like ATPases. Extraordinarily, sensitivity to CHD4 depletion seems to be specific to cancer cells as we and others observed that silencing of this chromatin remodeler had no influence on proliferation of human myoblasts, fibroblasts (*Böhm et al., 2016*), non-transformed mammary epithelial cells (MCF10A) (*D'Alesio et al., 2016*), and normal primary hematopoietic cells (*Heshmati et al., 2016*). We believe that this broad tumor dependency on CHD4 might be partially explained by its positive contribution to the activity of oncogenic transcription factors at super-enhancers. Besides FP-RMS, in glioblastoma, CHD4 co-localizes with the transcription factor ZFHX4 and co-regulates a subset of its target genes to maintain the tumor-initiating cell population (*Chudnovsky et al., 2014*).

Nonetheless, the role of CHD4 in DNA-damage repair and genome integrity (*Qi et al., 2016*; *Smeenk et al., 2010*) should be further explored in the context of general tumor sensitivity.

In conclusion, our data reveal CHD4 as a prominent and promising new target for SE-disruption therapy with a broad range of application. Its activity in the regulation of gene expression seems to be complex, involve both activation and repression of transcription and dependent on a variety of interaction partners whose function still requires clarification. We hope that our work stimulates the development of a CHD4-specific inhibitor which would allow further studies regarding the biological activity of this chromatin remodeler and the future assessment of its potential as a new target for cancer therapy.

# Materials and methods

## Key resources table

| Reagent type (species) or resource | Designation | Source or reference | Identifiers | Additional information |
|---|---|---|---|---|
| Cell line (*Homo-sapiens*) | RH4 (fusion-positive rhabdomyosarcoma) | Other | RRID:CVCL_5916 | See Materials and methods |
| Cell line (*Homo-sapiens*) | RH5 (fusion-positive rhabdomyosarcoma) | Other | RRID:CVCL_5917 | See Materials and methods |
| Cell line (*Homo-sapiens*) | SCMC (fusion-positive rhabdomyosarcoma) | Other | | See Materials and methods |
| Recombinant DNA reagent | lentiCRISPRv2 puro (plasmid) | Addgene | #98290; RRID:Addgene_98290 | Cas9 lentiviral expression construct |
| Recombinant DNA reagent | pU6-gRNA-EF1a-RFP657/BFP/EGFP (plasmid) | Other | | See Materials and methods |
| Recombinant DNA reagent | pRSIT-U6Tet-shRNA-PGKTetRep-2A-GFP-2A-puro (plasmid) | Cellecta Inc | Custom made | shRNA lentiviral expression construct, |
| Recombinant DNA reagent | PX459; pSpCas9(BB)−2A-Puro (plasmid) | Addgene | #62988; RRID:Addgene_ 62988 | Cas9 and sgRNA expression construct |
| Antibody | Recombinant Anti-Brd4 (rabbit monoclonal) | Abcam | #ab128874; RRID:AB_11145462 | WB (1:1000) |
| Antibody | BRD4 (rabbit polyclonal) | Bethyl Laboratories | #A301-985A100; RRID:AB_2620184 | ChIP (10 μg) |
| Antibody | Cas9 (mouse monoclonal) | Cell Signaling Technologies | CST:7A9-3A3; #14697; RRID:AB_2750916 | WB (1:1000) |
| Antibody | CHD4 (rabbit polyclonal) | Bethyl Laboratories | #A301-082A; RRID:AB_873002 | WB (1:1000) |
| Antibody | CHD4 (rabbit polyclonal) | Invitrogen | #PA5-27472; RRID:AB_2544948 | ChIP (10 μg) |
| Antibody | Anti-Flag (mouse monoclonal) | Sigma Aldrich | Sigma:M2; #F1804; RRID:AB_262044 | WB (1:1000), ChIP (10 μg), IF (1:250), IP (8 μg) |
| Antibody | FKHR/FOXO1 (rabbit polyclonal) | Santa Cruz Biotechnology | St.Cruz:H-128; #sc-11350; RRID:AB_640607 | WB (1:1000) |
| Antibody | GAPDH (rabbit monoclonal) | Cell Signaling Technologies | CST:14C10; #2118L; RRID:AB_561053 | WB (1:1000) |

*Continued on next page*

Continued

| Reagent type (species) or resource | Designation | Source or reference | Identifiers | Additional information |
|---|---|---|---|---|
| Antibody | HDAC1 (mouse monoclonal) | Cell Signaling Technologies | CST:10E2; #5356; RRID:AB_10612242 | WB (1:1000) |
| Antibody | HDAC2 (mouse monoclonal) | Cell Signaling Technologies | CST:3F3; #5113S; RRID:AB_10624871 | WB (1:1000) |
| Antibody | HDAC2 (rabbit polyclonal) | Abcam | #Ab7029; RRID:AB_305706 | ChIP (14.6 µg) |
| Antibody | Histone H3K9ac (rat monoclonal) | Active Motif | #61663; RRID:AB_2793725 | ChIP (10 µg) |
| Antibody | Histone H3K9me1 (rabbit polyclonal) | Active Motif | #39887; RRID:AB_2793381 | ChIP (10 µg) |
| Antibody | Histone H3K9me3 (rabbit polyclonal) | Active Motif | #39765; RRID:AB_2793334 | ChIP (10 µg) |
| Antibody | Histone H3K27ac (rabbit polyclonal) | Active Motif | #39133; RRID:AB_2561016 | ChIP (7 µg) |
| Antibody | Anti-MTA2 (mouse monoclonal) | Sigma Aldrich | #M7569; RRID:AB_477237 | WB (1:1000) |
| Antibody | MTA2/PID (rabbit polyclonal) | Abcam | #ab8106; RRID:AB_306276 | ChIP (5 µg) |
| Antibody | PAX3-FOXO1 breakpoint specific (mouse monoclonal) | doi:10.1158/0008–5472.CAN-10–0582 | | ChIP (10 µg) |
| Antibody | RBBP4 (rabbit polyclonal) | Bethyl Laboratories | #A301-206A; RRID:AB_890631 | WB (1:1000) |
| Antibody | RBBP4 (rabbit polyclonal) | EpiGentek | #A-2703–050 | ChIP (10 µg) |
| Antibody | RNA Pol II (rat monoclonal) | Active Motif | #61667; RRID:AB_2687513 | ChIP (15 µg) |
| Antibody | Alexa Fluor 594 anti-mouse (goat polyclonal) | Thermo Fisher Scientific | #A11032; RRID:AB_2534091 | IF (1:200) |
| Antibody | Spike-in Antibody (rabbit, clonality not specified) | Active Motif | #61686 | ChIP (2 µl) |
| Sequence-based reagent | Guide RNAs used in CRISPR/Cas9 screen | Microsynth | sgRNAs | See *Supplementary file 1* |
| Sequence-based reagent | sg_NCHD4 | Microsynth | sgRNA | 5'GAGCGGAAGG GGATGGCGTC 3' |
| Sequence-based reagent | sg_CCHD4 | Microsynth | sgRNA | 5'TCTGCATCTTC ACTGCTGCT 3' |
| Sequence-based reagent | sg_NBRD4 | Microsynth | sgRNA | 5'ATGTCTGCGGA GAGCGGCCCTGG 3' |
| Sequence-based reagent | Donor DNA | IDT | cDNA | See *Supplementary file 1* |
| Sequence-based reagent | Primers for ChIP-qPCR | Microsynth | | See Materials and methods |
| Peptide, recombinant protein | 3xFlag peptide | Sigma-Aldrich | #F4799 | IP elution (200 µg/ml) |
| Commercial assay or kit | Cell Proliferation ELISA, BrdU kit | Roche | #11647229001 | |

*Continued*

| Reagent type (species) or resource | Designation | Source or reference | Identifiers | Additional information |
|---|---|---|---|---|
| Commercial assay or kit | Pierce BCA Protein Assay Kit | Thermo Fisher Scientific | #23227 | |
| Commercial assay or kit | RNeasy mini Kit | Qiagen | #74106 | |
| Commercial assay or kit | ChIP-IT High Sensitivity kit | Active Motif | #53040 | |
| Commercial assay or kit | iDeal ChIP-seq kit for Transcription Factors | Diagenode | #C01010055 | |
| Commercial assay or kit | TruSeq ChIP Library Preparation Kit | Illumina | #IP-202–1012 | |
| Commercial assay or kit | NextSeq500 High Output Kit v2 | Illumina | #FC-404–2005 | |
| Commercial assay or kit | TruSeq Stranded Total RNA Sample Preparation Kit | Illumina | #20020596 | |
| Chemical compound, drug | DNase I recombinant, RNase-free | Roche | #04716728001 | |
| Chemical compound, drug | 7-amino-actinomycinD | Invitrogen | #A1310 | |
| Chemical compound, drug | Cell Proliferation Reagent WST-1 | Roche | #5015944001 | |
| Chemical compound, drug | Crystal Violet | Sigma-Aldrich | #V5265 | |
| Chemical compound, drug | ChIP Cross-link Gold | Diagenode | #C01019027 | |
| Software, algorithm | ProteoWizard (version 3.0.7494) | http://proteowizard.sourceforge.net/projects.html | RRID:SCR_012056 | |
| Software, algorithm | Trans-Proteomic Pipeline | doi:10.1002/pmic.200900375 | | |
| Software, algorithm | CRAPome 2.0 | doi:10.1038/nmeth.2557 | | |
| Software, algorithm | SAINTexpress | doi:10.1016/j.jprot.2013.10.023 | RRID:SCR_018562 | |
| Software, algorithm | BioGRID 3.5 | doi:10.1093/nar/gky1079 | RRID:SCR_007393 | |
| Software, algorithm | BWA | doi:10.1186/gb-2009-10-3-r25 | RRID:SCR_005476 | |
| Software, algorithm | igvtools | doi:10.1038/nbt.1754 | | |
| Software, algorithm | MACS2 | doi:10.1186/gb-2008-9-9-r137 | RRID:SCR_013291 | |
| Software, algorithm | BEDTools | doi:10.1093/bioinformatics/btq033 | RRID:SCR_006646 | |
| Software, algorithm | HOMER | doi:10.1016/j.molcel.2010.05.004 | RRID:SCR_010881 | |

*Continued on next page*

*Continued*

| Reagent type (species) or resource | Designation | Source or reference | Identifiers | Additional information |
|---|---|---|---|---|
| Software, algorithm | NGSplot | doi:10.1186/1471-2164-15-284 | RRID:SCR_011795 | |
| Software, algorithm | FastQC v0.11.7 | http://www.bioinformatics.babraham.ac.uk/projects/fastqc | RRID:SCR_014583 | |
| Software, algorithm | Hisat2 v2.1.0 | doi:10.1038/nmeth.3317 | RRID:SCR_015530 | |
| Software, algorithm | Samtools v1.7 | doi:10.1093/bioinformatics/btp352 | RRID:SCR_002105 | |
| Software, algorithm | QualiMap | doi:10.1093/bioinformatics/bts503 | RRID:SCR_001209 | |
| Software, algorithm | featureCounts v1.6.0 | doi:10.1093/bioinformatics/btt656 | RRID:SCR_012919 | |
| Software, algorithm | DESeq2 v3.7 | doi:10.1186/s13059-014-0550-8 | RRID:SCR_015687 | |
| Software, algorithm | GSEA 3.0 | doi:10.1073/pnas.050658010 | RRID:SCR_003199 | |
| Other | Spike-in Chromatin | Active Motif | #53083 | |

## Cell lines

The FP-RMS cell lines RH4 and RH5 were kindly provided by Dr. Peter Houghton (Greehey Children's Cancer Research Institute, San Antonio TX), and the SCMC cell line by Dr. Janet Shipley (The Institute of Cancer Research, London, United Kingdom). The HEK293T cell line was purchased from ATCC and used for virus production. All cell lines were routinely maintained in Dulbecco's modified Eagle's medium (Sigma-Aldrich) supplemented with 10% fetal bovine serum (Sigma-Aldrich), 1% GlutaMAX (Gibco) and 100 U/ml penicillin/streptomycin (Gibco), and cultured in 5% $CO_2$ at 37°C. The RH4, RH5 and SCMC cells were tested and authenticated by cell line typing analysis (STR profiling) in 2020 and positively matched (*Hinson et al., 2013*). All cells were routinely tested for *Mycoplasma*.

## CRISPR/Cas9 screen

RH4 cells stably expressing Cas9 were obtained by transducing wildtype cells with the expression vector lentiCRISPRv2 puro (#98290, Addgene) followed by puromycin selection (1 µg/mL). Guide RNAs targeting the NuRD subunits (five sgRNAs/subunit) were cloned into the RFP-labelled lentiviral sgRNA expression construct (pU6-gRNA-EF1a-RFP657), and the control guide targeting the AAVS1 locus into a similar BFP-labelled sgRNA expression construct. Both sgRNA expression vectors were kindly provided by Dr. Yun Huang, University Children's Hospital Zurich. Viruses were produced in HEK293T cells by co-transfection of pVSV-G, PAX2, and the sgRNA expression vector using $CaPO_4$. Medium was replaced 24 hr after transfection and viruses were harvested after additional 48 hr. Viral supernatant was cleared by centrifugation, filtered, and concentrated (Amicon Ultra, 100 KDa, 15 mL, Millipore). Then, Cas9 expressing RH4 cells were infected, in the presence of 8 µg/ml polybrene, with viral supernatant containing either RFP- or BFP-labelled sgRNA expression vectors. Two days after transduction the RFP and BFP populations were mixed 1:1. The distribution of RFP and BFP populations was assessed by flow cytometry at day 2 and 12 after transduction. Double knockouts were performed by cloning guide RNAs targeting one of the NuRD member paralogs (MBD3, HDAC2 and GATAD2B) into an EGFP-labelled lentiviral sgRNA expression construct (pU6-gRNA-EF1a-EGFP). Then, Cas9 expressing RH4 cells were infected with viral supernatant containing RFP- and EGFP-labelled sgRNA expression vectors targeting both NuRD paralogs. Two days after transduction the RFP/EGFP population was mixed 1:1 with the BFP-labelled control population. The distribution of the double positive RFP and EGFP population was assessed by flow cytometry at day 2 and 12 after transduction and compared with the single positive BFP control population. One technical replicate and three biological replicates were performed per sample. Biological replicates are

tests performed on biologically distinct samples representing an identical time point or treatment dose while technical replicates are tests performed on the same sample multiple times. The guide RNAs sequences are displayed in *Supplementary file 1*.

## Doxycycline-inducible knockdowns

Two pRSIT-U6Tet-shRNA-PGKTetRep-2A-GFP-2A-puro vectors containing shRNAs targeting RBBP4 were purchased from Cellecta Inc with the following sequences: shRBBP4#1 5' GCCTTTCTTTCAA TCCTTATA 3' and shRBBP4#2 5' ATGAACCTTGGGTGATTTGTT 3'. Viruses were produced by co-transfecting the shRNA expression vectors together with the lentiviral packaging and envelope plasmids (pMDL, pREV, and pVSV-G, kindly provided by Oliver Pertz, Department of Biomedicine, University of Basel, Switzerland) into HEK293T cells using $CaPO_4$. Medium was replaced 24 hr after transfection and viruses were harvested after additional 48 hr. Viral supernatant was cleared by centrifugation, filtered, and concentrated (Amicon Ultra, 100 KDa, 15 mL, Millipore). Then, wildtype RH4 cells were transduced with the concentrated viral particles for 24 hr in the presence of 8 µg/ml polybrene and selected with puromycin (1 µg/mL). ShRNA expression was induced using 100 ng/ml doxycycline.

## Cell viability and death assays

Cells were cultured in a 96-well format, treated with doxycycline and, at various time points, their viability was measured by WST-1 assay, crystal violet and BrdU staining. For the WST-1 assay, cells were incubated with the Cell Proliferation Reagent WST-1 (Cat#5015944001, Roche) for at least 20 min and absorbance was measured in a plate reader at 640 nm and 440 nm. For the crystal violet assay, cells were first fixed with 4% paraformaldehyde and then stained with a 0.05% crystal violet solution (Sigma-Aldrich) which was later dissolved in methanol. Absorbance was measured by a plate reader at 540 nm. BrdU incorporation was determined using the Cell Proliferation ELISA, BrdU kit (Roche) according to manufacturer's recommendations. Results were normalized to untreated cells and three biological and technical replicates were performed for each experiment. Cell death was assessed by 7-AAD staining (7-amino-actinomycin D). For this assay, cells were grown in the 6-well plate format and collected at various time points after shRNA expression induction by doxycycline treatment and stained with 7-AAD (Invitrogen) at a 1:500 dilution. The percentage of dead cells was measured by flow cytometry. Results were normalized to untreated cells and 1 technical and three biological replicates were performed.

## CRISPR/Cas9 Flag knockin

The knockin of a 3xFlag tag into endogenous *CHD4* or *BRD4* in RH4 cells was performed by CRISPR/Cas9-mediated homologous repair (*Ran et al., 2013*). Guide RNAs targeting the N- or C-terminus of *CHD4* or the N-terminus of *BRD4* (sg_NCHD4: 5' GAGCGGAAGGGGATGGCGTC 3', sg_CCHD4: 5' TCTGCATCTTCACTGCTGCT 3', sg_NBRD4: 5' ATGTCTGCGGAGAGCGGCCCTGG 3') were cloned into the pSpCas9(BB)−2A-Puro PX459 vector (#62988, Addgene). Then, the PX459 vector was transiently co-transfected with the respective donor DNAs (*Supplementary file 1*) into RH4 cells using JetPrime reagent (Polyplus Transfections). One day after transfection, cells were incubated with 1 µM of non-homologous end-joining inhibitor SCR7. Clones were obtained by limited dilution. Flag insertion was confirmed by immunofluorescence. For that, cells were seeded on cover slides, fixed with 4% paraformaldehyde for 15 min, permeabilized for 15 min with 0.1% Triton X-100 in PBS and blocked with 4% horse serum in 0.1% Triton X-100 in PBS. All steps were carried out at room temperature. Then, cells were incubated overnight with anti-Flag antibody (clone M2, #F1804, Sigma-Aldrich) at a 1:250 dilution. Fluorescent secondary antibody (Alexa Fluor 594 anti-mouse, #A11032, Thermo Fisher Scientific), at a 1:200 dilution, was applied for 1 hr at room temperature. Cover slides were fixed on objective glass with DAPI Vectashield mounting medium (Vector laboratories Inc) and analyzed by fluorescence microscopy. Knockin cells were also evaluated for changes in proliferation rate by counting RH4 cells with or without Flag knockin every day for 6 days. Three biological and two technical replicates were performed.

## Flag immunoprecipitation

RH4 cells expressing endogenous 3xFlag tagged CHD4 (both N- and C-terminus) or 3xFlag tagged BRD4 (N-terminus) were grown to confluency in 15 cm dishes. Per condition, three confluent dishes were used. Cells were washed with PBS, harvested, and lysed in sucrose buffer (320 mM sucrose, 3 mM $CaCl_2$, 2 mM MgOAc, 0.1 mM EDTA, 10 mM DTT, 0.5 mM PMSF, 0.25% NP-40). The nuclei were pelleted by centrifugation (10 min, 1100 g, 4°C) and lysed by incubation with lysis buffer (50 mM HEPES pH 7.8, 3 mM $MgCl_2$, 300 mM NaCl, 1 mM DTT, 0.1 mM PMSF) in the presence of 15 U/µl of benzonase for 1 hr at 4°C. Before antibody incubation, an input sample was collected and stored at −20°C. Protein G Dynabeads were coupled with 8 µg of anti-Flag antibody (clone M2, #F1804, Sigma-Aldrich) per plate and incubated overnight together with the nuclear extracts. After several washes, immunoprecipitates were eluted in elution buffer (50 mM Tris-HCl pH 7.4, 150 mM NaCl) supplemented with 200 µg/ml of 3xFlag peptide (Sigma-Aldrich). As negative control, RH4 wildtype cells were used. For all experiments, at least three biological replicates were performed.

## Liquid chromatography-mass spectrometry (LC-MS)

Samples were prepared and data were acquired at the Functional Genomics Center Zurich (FGCZ). Protein digestion was performed according to the filter-aided sample preparation method (FASP) (*Wiśniewski et al., 2009*). Briefly, proteins were loaded into filter units in the presence of buffer UA (8M urea in 100 mM Tris-HCl, pH 8.2) supplemented with 0.1M DTT. Then, the samples were washed with buffer UA, the IAA solution (0.05M iodoacetamide in buffer UA), and a 0.5M NaCl solution. Protein digestion with trypsin (1:50) was carried out overnight at room temperature in the presence of 0.05M triethylammonium carbonate. Eluted peptides were acidified with trifluoroacetic acid to a final concentration of 0.5%, purified with SPE C18 columns, and resolved in a 3% acetonitrile, 0.1% formic acid (FA) solution for mass spectrometry analysis. Dissolved samples were injected by an Easy-nLC 1000 system (Thermo Scientific) and separated on a self-made reverse-phase column (75 µm x 150 mm) packed with C18 material (ReproSil-Pur, C18, 120 Å, AQ, 1.9 µm, Dr. Maisch GmbH). The column was equilibrated with 100% solvent A (0.1% FA in water). Peptides were eluted using the following gradient of solvent B (0.1% FA in ACN): 0–70 min at 3–30% B followed by 70–75 min at 30–97% B with a flow rate of 0.3 µl/min. High accuracy mass spectra were acquired with an Orbitrap Fusion (Thermo Scientific) that was operated in data-dependent acquisition mode. All precursor signals were recorded in the Orbitrap using quadrupole transmission in the mass range of 300–1500 m/z. Spectra were recorded with a resolution of 120 000 at 200 m/z, a target value of 4E5 and a maximum cycle time of 3 s. Data-dependent MS/MS were recorded in the linear ion trap using quadrupole isolation with a window of 1.6 Da and HCD fragmentation with 35% fragmentation energy. The ion trap was operated in rapid scan mode with a target value of 2E3 and a maximum injection time of 300 ms. Precursor signals were selected for fragmentation with a charge state from +two to +seven and a signal intensity of at least 5E3. A dynamic exclusion list was used for 30 s and maximum parallelizing ion injections was activated. Then, MS raw data were converted using ProteoWizard (*Kessner et al., 2008*) (version 3.0.7494) to mzXML profile files, which were searched for trypsin cleavage of specific peptides using the search engines X! TANDEM Jackhammer TPP (2013.06.15.1 - LabKey, Insilicos, ISB), omssacl (*Geer et al., 2004*) (version 2.1.9), MyriMatch (*Tabb et al., 2007*) 2.1.138 (2012-12-1), and Comet (*Eng et al., 2013*) (version 2016.01 rev. 3) with a 10 ppm peptide precursor mass error and 0.4 Da fragment mass error, against a non-redundant canonical reviewed *Homo sapiens* protein database obtained from uniProtKB/Swiss-Prot (downloaded on 2019.04.01). The protein database was appended with decoys by reverting the original protein sequence. Carbamidomethylation on cysteine residues was set as static modification and two missed cleavages were allowed. To control for false identifications, peptides were analyzed with the Trans-Proteomic Pipeline (*Deutsch et al., 2010*) (TPP v4.7 POLAR VORTEX rev 0, Build 201403121010), using PeptideProphet, iProphet, and ProteinProphet scoring (*Choi et al., 2008*). The identified protein spectral counts and peptides for ProteinProphet were filtered at a 0.01 FDR using myu-proFDR (*Reiter et al., 2009*), corresponding to a 0.996725 iprophet probability.

## LC-MS analysis

The data were median normalized, and proteins only considered for subsequent probabilistic scoring if they were detected in two out of three replicates per condition. The remaining 416 proteins were

submitted as Spectral Count table to CRAPome (*Mellacheruvu et al., 2013*) 2.0 for SAINT scoring with SAINTexpress (*Teo et al., 2014*). The probabilistic SAINT Score was calculated using default SAINTexpress options and additional 30 FLAG-tagged CRAPome controls, originating from various cellular backgrounds (e.g. HeLa, Hek293 and U-2 OS cells). For calculating the primary fold change (FCA), all negative controls, including the experimental and the appended CRAPome controls, were used. Scored proteins were then categorized into high, medium, and low confidence interactors by FCA, Saint Probability Score (SPC), and average spectral counts (avg. SC): high confidence - FCA $\geq$4, SPC $\geq$ 0.99, and avg. SC >3; medium confidence - FCA $\geq$3, SPC $\geq$ 0.90, and avg. SC >3; and low confidence - FCA $\geq$2, SPC $\geq$ 0.6, and avg. SC >3. This filtering resulted in 100 significant interaction partners for the N-terminus tagged CHD4, and 82 for the C-terminus tagged CHD4. Interaction partners were compared with previous reports available in BioGRID 3.5 (downloaded on the 28.08.2019, *Homo sapiens* subset). The result of this analysis is available on *Figure 2—source data 1*.

## Western blot

Cells were lysed in standard lysis buffer (50 mM Tris-HCl pH 7.5, 150 mM NaCl, 1% NP-40, 0.5% sodium deoxycholate, 0.1% SDS, 1 mM EGTA, 50 mM NaF, 5 mM $Na_4P_2O_7$, 1 mM $Na_3VO_4$, and 10 mM ß-glycerol phosphate in the presence of protease inhibitors, cOmplete Mini, Roche). Protein concentration was measured using the Pierce BCA Protein Assay Kit (Thermo Fisher Scientific) according to the manufacturer's instructions. Then, proteins were separated using NuPAGE 4–12% Bis-Tris pre-cast gels (Thermo Fisher Scientific) and transferred into nitrocellulose membranes (GE Healthcare). Membranes were blocked with 5% milk in TBST, incubated overnight at 4°C with primary antibodies diluted 1:1000, and then incubated for 1 hr with HRP-linked secondary antibodies at room temperature. Finally, proteins were detected by chemiluminescence. The following primary antibodies were used: BRD4 (ab128874, Abcam), Cas9 (7A9-3A3, 14697, Cell Signaling Technologies), CHD4 (A301-082A, Bethyl Laboratories), Flag (clone M2, F1804, Sigma Aldrich), FOXO1/FKHR (H-128, sc-11350, Santa Cruz Biotechnology), GAPDH (14C10, 2118L, Cell Signaling Technologies), HDAC1 (10E2, 5356, Cell Signaling Technologies), HDAC2 (3F3, 5113S, Cell Signaling Technologies), MTA2 (M7569, Sigma Aldrich), and RBBP4 (A301-206A-M, Bethyl Laboratories).

## Quantitative real time PCR

Total RNA was extracted using the RNeasy mini Kit (Qiagen Instruments AG) and cDNA synthesis was carried out using the High-Capacity Reverse Transcription Kit (Applied Biosystems by Thermo Fisher Scientific), according to manufacturer's instructions. Quantitative PCR was performed using TaqMan gene expression master mix (Applied Biosystems) and TaqMan gene expression assays (Applied Biosystems). Data were analyzed with the SDS 2.3 software and Ct values were normalized to GAPDH. Relative expression levels were calculated using the $\Delta\Delta$Ct method based on experiments performed in triplicates (3 biological and three technical replicates). Outliers found in the technical replicates (SD >0.5) were removed from the analysis. The following TaqMan gene expression assays were used: ALK (Hs00608284_m1), ASS1 (Hs01597981_gH), CDH3 (Hs01285856_cn), CHD4 (Hs00172349_m1), CNR1 (Hs01038522_s1), GAPDH (Hs02758991_g1), PAX3-FOXO1 (Hs03024825_ft), PIPOX (Hs04188864_m1), RBBP4 (Hs01568507_g1), and TFAP2B (Hs00231468_m1).

## Chromatin immunoprecipitation

ChIP assays were performed using the ChIP-IT High Sensitivity kit (Active Motif) according to the manufacturer's instructions. Briefly, cells were grown to confluence in 15 cm dishes, fixed with 1% formaldehyde for 13 min, harvested and sonicated with the EpiShear ProbeSonicator (Active Motif) for 27 cycles (30% amp, 30 s ON, 30 s OFF). ChIP assays performed in RH5 and SCMC cells as well as for MTA2 in RH4 cells were done using the iDeal ChIP-seq kit for Transcription Factors (#C01010055, Diagenode) according to the manufacturer's instructions. For these assays, cells were fixed with ChIP Cross-link Gold for 30 min (#C01019027, Diagenode) and 1.1% formaldehyde for 15 min, and the chromatin was sheared with the Bioruptor Pico sonication device (#B01080010, Diagenode) for 5 to 10 cycles. Sonicated lysates were then quantified and 30 µg of chromatin were incubated overnight at 4°C with 5–15 µg of antibody. DNA was purified according to the manufacturer's

instructions. Quantitative PCR was performed using PowerUp SYBR Green Master Mix (Thermo-Fisher Scientific AG) and primers were designed to target known binding sites of PAX3-FOXO1. The untranscribed genomic region (*UNTR5*) was used as negative control and the commercially available Human Negative Control Primer Set 1 (#71001, Active Motif) was used for normalization. For ChIP-qPCR, three technical replicates and two biological replicates were performed. The following anti-bodies were used for ChIP: BRD4 (A301-985A100, Bethyl Laboratories), Flag (clone M2, F1804, Sigma Aldrich), H3K9ac (61663, Active Motif), H3K9me1 (39887, Active Motif), H3K9me3 (39765, Active Motif), H3K27ac (39133, Active Motif), HDAC2 (Ab7029, Abcam), MTA2 (Ab8106, Abcam), RBBP4 (A-2703–050, Epigentek), and RNA polymerase 2 (61667, Active Motif). The following primer sequences were used for ChIP-qPCR:

> ALK_FW: 5' GTCACTTTGGGTCACTTGCT 3'
> ALK_RV: 5' GCCTTGTAGTTAGCTCTCCC 3'
> ASS1_FW: 5' CAATGGTGGAGCGTGAAAT 3'
> ASS1_RV: 5' ACCCTCCCATTCTCTTTGC 3'
> CDH3_FW: 5' ATGCTCCCGAGATACCAGAT 3'
> CDH3_RV: 5' AGAAGCGTTGTAATCCTCCAA 3'
> UNTR5_FW: 5' TATAAAGGACCGTGGCTTCC 3'
> UNTR5_RV: 5' TCATTCATTTGGTCATGGCT 3'

## DNase I hypersensitivity assays

DNase I hypersensitivity assays were performed as previously described (*Jin et al., 2015*). First, cells were washed and resuspended in RSB buffer (10 mM Tris-HCl pH 7.4, 10 mM NaCl, 3 mM MgCl$_2$, 0.2% Triton X-100). Then, DNase I (10 µL of a 0.33 U/µL DNase solution in RSB buffer) was added to a total of 40 µL of cells (containing 10,000 cells) followed by a 5 min incubation at 37 ˚C. The digestion was halted with 50 µL of stop buffer (10 µM Tris-HCl pH 7.4, 10 µM NaCl, 10 µM EDTA, 0.15% SDS, supplemented with 125 µL of proteinase K per 10 mL). Proteinase K activation at 55 ˚C for 1 hr was followed by DNA purification (NucleoSpin Gel and PCR Clean-up, Macherey-Nagel GmbH and Co. KG). Library preparation was performed as for ChIP-seq samples, except that paired-end was employed rather than single-end sequencing on the NextSeq 500 (Illumina).

## Library preparation for ChIP-seq assays

DNA libraries were created using Illumina TruSeq ChIP Library Prep Kit (Cat#IP-202–1012), after which DNA was size selected with SPRI select reagent kit (to obtain a 250–300 bp average insert fragment size). Then, libraries were multiplexed and sequenced using NextSeq500 High Output Kit v2 (75 cycles, #FC-404–2005) on an Illumina NextSeq500 machine. 25,000,000–30,000,000 unique reads were generated per sample.

## ChIP-seq data processing, peak calling, and annotation

ChIP enriched DNA reads were mapped to reference genome (version hg19) using BWA (*Langmead et al., 2009*). Duplicate reads were not discarded. For IGV sample track visualization, coverage density maps (tdf files) were generated by extending reads to the average size (measured by Agilent Bioanalyzer minus 121 bp for sequencing adapters) and counting the number of mapped reads to 25 bp windows using igvtools (*Robinson et al., 2011*) (https://www.broadinstitute.org/igv/igvtools). ChIP-seq read density values were normalized to million mapped reads. Then, high-confidence ChIP-seq peaks were called by MACS2 (https://github.com/taoliu/MACS) with the narrow algorithm (*Zhang et al., 2008*). The peaks which overlapped with the possible anomalous artifact regions (such as high-mappability regions or satellite repeats) blacklisted by the ENCODE consortium (https://sites.google.com/site/anshulkundaje/projects/blacklists) were removed using BEDTools (*Quinlan and Hall, 2010*). Peaks from ChIP-seq were selected at a stringent p-value of at least 0.0000001. Peaks within 2,500 bp to the nearest TSS were set as promoter proximal, while all other were considered distal. The distribution of peaks (as intronic, intergenic, etc.) was annotated using HOMER (*Heinz et al., 2010*). Metagene plots and heatmaps of ChIP-seq and DNase I hypersensitivity data were obtained with NGSplot (*Shen et al., 2014*). Scaling and plot adjustments were performed using the replot.r function (https://github.com/shenlab-sinai/ngsplot/wiki/ProgramArguments101; *Shen et al., 2014*). Colors for heatmaps were set in Adobe Photoshop,

while sizing and placement were performed in Adobe Illustrator. Overlaps between ChIP-seq peaks were calculated using BEDTools intersect and displayed as area-weighted Venn diagrams created using the R Vennerable package. The previously established Enhancer Domain Expression Nexus (EDEN) (*Gryder et al., 2017*) script was used to annotate and link ChIP-seq peaks at enhancers and super-enhancers to their putative target genes. Enhancer-gene connections were restricted to TAD domains (*Dixon et al., 2012*), to genes with expression level of TPM >1 and to genes within 500,000 bp from enhancers. One gene upstream, one gene downstream and any gene overlapping (often intronic regulatory elements were found) were considered as putative targets for a given ChIP-seq peak.

## Correlation matrix

To correlate all ChIP-seq experiments performed in RH4 cells, we began by creating a coordinate BED file that combined and merged all called peaks from selected ChIP-seq and DNase data files. Then, the read counts for each experiment across the combined BED were calculated, from which the pair-wise Pearson correlation was calculated across all possible 2-sample comparisons. These Pearson correlations where then used to cluster an all-by-all matrix and to group epigenomic entities by the similarity of their genome-wide profiles.

## Chromatin states mapping

Chromatin states were identified using the hidden Markov model to find complex patterns of chromatin modifications (http://compbio.mit.edu/ChromHMM/) (*Ernst et al., 2011*; *Ernst and Kellis, 2012*). We defined chromatin states by integrated analysis of 9 histone modifications (H3K27ac, H3K27me3, H3K4me1, H3K4me2, H3K4me3, H3K36me3, H3K9ac, H3K9me1, and H3K9me3) and two architectural proteins (RAD21 and CTCF), allowing us to identify 16 states in RH4 cells.

## Spike-in analysis

ChIP-Rx assays, or ChIP-seq assays with reference exogenous chromatin, were performed with the shRNA cell lines (with the purpose of comparing wildtype RH4 cells expressing a scramble construct with CHD4 knockdown). Briefly, chromatin from RH4 shScramble and shCHD4 cells after 48 hr of doxycycline treatment was collected and spiked-in with *Drosophila* chromatin (Spike-in chromatin, Cat#53083, Active Motif) and an antibody against the *Drosophila* specific histone variant H2Av (Spike-in antibody, Cat# 61686, Active Motif). As these agents are introduced at identical amounts and concentrations during the ChIP reactions, technical variation associated with downstream steps is accounted for. Normalization is then achieved by calculating reads per million mapped *Drosophila* reads, as previously described (*Orlando et al., 2014*).

## Super-enhancer identification

Super-enhancers were identified as previously described (*Gryder et al., 2017*). Briefly, RH4 enhancers were identified using the ROSE2 (Rank Order of Super Enhancers) software and H3K27ac ChIP-seq peaks distal to TSSs (>2,500 bp from TSS). Enhancer constituents were stitched together if clustered within 12.5 kb. Enhancers were classified into typical and super-enhancers (TEs and SEs) based on a cutoff at the inflection point in the rank ordered set (where tangent slope = 1) of the ChIP-seq signal (input normalized). A list of all SEs identified is found on *Figure 4—source data 1*.

## RNA-seq sample and library preparation

RNA was isolated 24 and 48 hr after the start of doxycycline treatment using the RNeasy Mini Kit (Qiagen), including the DNase digestion step, according to the manufacturer's instructions. Library preparation and sequencing was performed by ATLAS Biolabs GmbH, Berlin, Germany. Briefly, cDNA libraries were prepared with the Illumina TruSeq Stranded Total RNA Sample Preparation Kit including Ribo-Zero (Cat#20020596)and paired-end sequenced on an Illumina HiSeq4000. For each assay, three biological replicates were performed.

## RNA-seq data analysis

Illumina adaptors were trimmed off and quality control was performed with FastQC v0.11.7 (http://www.bioinformatics.babraham.ac.uk/projects/fastqc). Then, the paired-end RNA-seq reads were

aligned to the GRCh38 reference genome (ftp://ftp.ensembl.org/pub/release95/fasta/homo_sapiens/dna/Homo_sapiens.GRCh38.dna_sm.primary_assembly.fa.gz) using Hisat2 v2.1.0 (*Kim et al., 2015*) with the options '–rna-strandness RF –fr' and Samtools v1.7 (*Li et al., 2009*). Mapping quality was assessed by QualiMap v2.2.1 (*García-Alcalde et al., 2012*). Read counts were measured at the gene level and annotated according to Ensembl95 (ftp://ftp.ensembl.org/pub/release95/gtf/homo_sapiens/Homo_sapiens.GRCh38.91.gtf.gz), using featureCounts v1.6.0 with the options '-p -B -O -M –fraction' (*Liao et al., 2014*). Genes with less than a sum of 10 reads were filtered out and differential gene expression analysis was conducted in RStudio v3.4.3 using DESeq2 v3.7 (*Love et al., 2014*). Significant differential gene expression was defined by fold change ≥25% and false discovery rate ≤0.01. Gene expression was clustered using the Pearson correlation method and heatmaps were obtained with the R package pheatmap. Enrichment for curated gene sets (Molecular Signatures Database v6.2) was performed using GSEA software version 3.0 (*Subramanian et al., 2005*) (21). The differentially expressed gene lists were pre-ranked by the metric value calculated as log10(p-value) divided by the reverse sign of log2(fold-change). The GSEA Preranked tool was employed with 1000 permutations, and only gene sets with a maximum list size of 500 and a minimum of 15 were considered.

## Analysis of the genome-wide cancer genetic vulnerability screens

The two datasets used, Combined RNAi and CRISPR (Avana) Public 19Q2, were download from the depmap online platform (https://depmap.org/portal/download/). Then, the sensitivity scores for NuRD subunits and SNF2-like ATPases were plotted in RStudio v3.4.3 using ggplot2.

## Statistics

All statistical analysis (except for sequencing and mass spectrometry data) were performed with the GraphPad prism software, version 8. Data are represented as mean ± SD, unless otherwise noted in the figure legend. Statistical significance was calculated by the ratio paired t test or one-way ANOVA and n represents number of biological replicates.

## Availability of data and materials

The proteomics dataset supporting the conclusions of this article is available in the ProteomeXchange Consortium via the PRIDE (*Perez-Riverol et al., 2019*) repository with the dataset identifier PXD015231. High-throughput ChIP-seq and DNase data are available through Gene Expression Omnibus (GEO) Superseries with the accession numbers GSE140115 and GSE155861. ChIP-seq data for H3K27ac, H3K27me3, H3K36me3, H3K4me1, H3K4me2, H3K4me3, BRD4, CTCF, RAD21, HDAC2, and RNA Polymerase 2 as well as DNase I hypersensitivity data obtained for wildtype RH4 cells were previously published (*Gryder et al., 2019b*; *Gryder et al., 2017*) and are available on the same data repository with the gene accession numbers GSE83728 and GSE116344. The RNA-seq data are available in the European Nucleotide Archive (ENA) with the accession number PRJEB34220. This study did not generate new code.

## Acknowledgements

We are very grateful to Dr. Marielle Yohe at National Cancer Institute-NIH, USA, for her support and helpful discussions, to Dr. Silvia Pomella at the Ospedale Pediatrico Bambino Gesú IRCCS Rome, Italy, for her practical assistance, and to Dr. Philippe Jacquet and Dr. Nicolas Salamin at the Center for Advanced Modeling Science (CADMOS) in Lausanne, Switzerland, for the fruitful collaboration and supportive Vital-IT access to the high performing computing resources in establishing our RNA-seq analysis pipeline. Also, we would like to thank Dr. Yun Huang at the University Children's Hospital Zurich, Switzerland, for providing us with the sgRNA expression vector.

## Additional information

### Funding

| Funder | Grant reference number | Author |
|---|---|---|
| Swiss National Science Foundation | 310030_156923 | Beat W Schäfer |
| Cancer League Switzerland | KLS-3868-02-2016 | Beat W Schäfer |
| Childhood Cancer Research Foundation Switzerland | | Beat W Schäfer |
| Innovative Medicines Initiative ULTRA-DD | 115766 | Fabian Frommelt Matthias Gstaiger |
| Swiss National Science Foundation | 31003A_175558 | Beat W Schäfer |

The funders had no role in study design, data collection and interpretation, or the decision to submit the work for publication.

### Author contributions

Joana G Marques, Conceptualization, Data curation, Formal analysis, Validation, Investigation, Visualization, Methodology, Writing - original draft, Project administration; Berkley E Gryder, Conceptualization, Data curation, Software, Formal analysis, Writing - review and editing; Blaz Pavlovic, Yeonjoo Chung, Validation, Investigation; Quy A Ngo, Fabian Frommelt, Software, Formal analysis; Matthias Gstaiger, Resources; Young Song, Katharina Benischke, Dominik Laubscher, Investigation; Marco Wachtel, Conceptualization, Supervision, Writing - review and editing; Javed Khan, Conceptualization, Resources, Supervision, Writing - review and editing; Beat W Schäfer, Conceptualization, Supervision, Funding acquisition, Writing - review and editing

### Author ORCIDs

Joana G Marques (iD) https://orcid.org/0000-0001-7152-9655
Fabian Frommelt (iD) http://orcid.org/0000-0003-3666-8005
Beat W Schäfer (iD) https://orcid.org/0000-0001-5988-2915

### Decision letter and Author response

Decision letter https://doi.org/10.7554/eLife.54993.sa1
Author response https://doi.org/10.7554/eLife.54993.sa2

## Additional files

### Supplementary files

• Supplementary file 1. Sequence of guide RNAs used for the NuRD-centered CRISPR screen and donor DNA sequences used in the CRISPR/Cas9-mediated Flag knockins.

• Transparent reporting form

### Data availability

The proteomics dataset supporting the conclusions of this article is available in the ProteomeXchange Consortium via the PRIDE (Perez-Riverol et al., 2019) repository with the dataset identifier PXD015231. High-throughput ChIP-seq and DNase data are available through Gene Expression Omnibus (GEO) Superseries with the accession numbers GSE140115 and GSE155861. ChIP-seq data for H3K27ac, H3K27me3, H3K36me3, H3K4me1, H3K4me2, H3K4me3, BRD4, CTCF, RAD21, HDAC2, and RNA Polymerase 2 as well as DNase I hypersensitivity data obtained for wildtype RH4 cells were previously published (Gryder et al., 2019b, 2017) and are available on the same data repository with the gene accession numbers GSE83728 and GSE116344. The RNA-seq data is available in the European Nucleotide Archive (ENA) with the accession number PRJEB34220.

The following datasets were generated:

| Author(s) | Year | Dataset title | Dataset URL | Database and Identifier |
|-----------|------|---------------|-------------|-------------------------|
| Gryder BE, Wen X, Khan J | 2019 | CHD4 regulates super-enhancer accessibility in fusion-positive rhabdomyosarcoma and is essential for tumor | https://www.ncbi.nlm.nih.gov/geo/query/acc.cgi?acc=GSE140115 | NCBI Gene Expression Omnibus, GSE140115 |
| Gryder BE, Wen X, Khan J | 2020 | NuRD subunit CHD4 regulates super-enhancer accessibility in Rhabdomyosarcoma and represents a general tumor dependency | https://www.ncbi.nlm.nih.gov/geo/query/acc.cgi?acc=GSE155861 | NCBI Gene Expression Omnibus, GSE155861 |

The following previously published datasets were used:

| Author(s) | Year | Dataset title | Dataset URL | Database and Identifier |
|-----------|------|---------------|-------------|-------------------------|
| Gryder BE, Yohe ME, Chou HC, Zhang X, Khan J | 2017 | Epigenetic Lanscape and BRD4 Transcriptional Dependency of PAX3-FOXO1 Driven Rhabdomyosarcoma | https://www.ncbi.nlm.nih.gov/geo/query/acc.cgi?acc=GSE83728 | NCBI Gene Expression Omnibus, GSE83728 |
| Gryder BE, Wen X, Khan J | 2019 | Selective Disruption of Core Regulatory Transcription [ChIP-seq] | https://www.ncbi.nlm.nih.gov/geo/query/acc.cgi?acc=GSE116344 | NCBI Gene Expression Omnibus, GSE116344 |

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
