## [Decision Letter]

Thank you for submitting your article "NuRD subunit CHD4 regulates super-enhancer accessibility in Rhabdomyosarcoma and represents a general tumor dependency" for consideration by *eLife*. Your article has been reviewed by three peer reviewers, including Xiaobing Shi as the Reviewing Editor and Reviewer #1, and the evaluation has been overseen by Kevin Struhl as the Senior Editor.

The reviewers have discussed the reviews with one another and the Reviewing Editor has drafted this decision to help you prepare a revised submission.

Summary:

The authors provide a combination of CRISPR/Cas9 knockout, co-immunoprecipitation, and ChIP-seq studies to argue that CHD4 is a potential therapeutic target for the treatment of PAX3-FOXO1 fusion-positive rhabdomyosarcoma. This work builds on the previous observation that CHD4 knockout decreases the growth of FP-RMS cell lines. The studies are fairly thorough and most of the experiments appear well-done. However, the primary conclusions that CHD4 functions independently of NuRD and that there exists a NuRD-only complex are not well supported due to several flaws in the approach and interpretation.

Essential revisions:

1) Only one cell line was used throughout the entire mechanistic study. Are there other PF-positive cell lines that the authors could use to validate at least some of the findings?

2) The authors perform a CRISPR/Cas9 screen of major NuRD components which shows that knockout of CHD4 and RBBP4 reduce viability of FP-RMS cells to a greater extent than other components. Based on this observation, they conclude that CHD4 functions independently of the full NuRD in FP-RMS (subsection “CHD4, unlike other NuRD members, is essential for FP-RMS cell viability”, first paragraph). However, there are multiple orthologs for all NuRD core components that may or may not functionally substitute for one another. As they demonstrate in the supplementary material, multiple orthologs are expressed in these cells (MBD2 and MBD3, MTA1 and MTA2). Therefore, although knockout MBD2 or MBD3 alone does not recapitulate the phenotype of CHD4 knockout, simultaneous knockout of both MBD2 and MBD3 indeed recapitulates the phenotype (personal communications). Hence, the conclusion that CHD4 is acting independently of NuRD is not correct. The authors need to either modify this conclusion or provide additional data by simultaneously eliminating functionally substituting orthologs (e.g. MBD2/MBD3 or MTA1/MTA2/MTA3).

3) The authors claim that a NuRD-only complex localizes to distinct regions (TSS) as compared to CHD4-NuRD. This claim is based on ChIP-seq of CHD4, HDAC2, and RBBP4 (subsection “CHD4/NuRD localizes to enhancers while CHD4-free NuRD to promoters”, last paragraph). However, as the authors later acknowledge (Discussion, second paragraph) RBBP4 and HDAC2 proteins are found in other chromatin-associated complexes (e.g. RBBP4 is found in PRC2, NuRF, and SIN3 complexes; HDAC2 is found in SIN3 and CoREST complexes). Hence, ChIP-seq of HDAC2 and RBBP4 does not necessarily reflect a NuRD-only complex and is inappropriate for this analysis and to make this claim. ChIP-seq of the MTA proteins would be much more appropriate for this experiment. As far as I am aware, the MTA proteins have not been found in other chromatin complexes; hence, they function as core NuRD components (Zhang et al., 2016) Therefore, the authors need to perform ChIP-seq for MTA proteins instead of HDAC2/RBBP4 in order to support this conclusion.

4) The authors endogenously Flag-tag the CHD4 protein to identify co-purifying proteins by mass spectrometry analyses. Based on this work, they develop a model in which CHD4 interacts with a different set of chromatin-associated proteins (BRD4). However, the key challenge with this approach is that CHD4 (NuRD) strongly interacts with chromatin. Hence, it is very difficult to determine whether co-purification reflects direct protein-protein interaction or indirect binding bridged by chromatin. While CHD4 itself has not been shown to directly bind DNA (as the authors point out), it does have chromatin-binding domains (PHD and chromodomains). Of note, histone proteins are among those identified in the mass-spectrometry data (Figure 2) indicating that NCPs are being co-purified. Furthermore, most of the co-purified proteins are from other chromatin-associated complexes or NuRD. These results suggest to me that the majority of the co-purified proteins could easily be explained by indirect/non-specific interaction through NCPs/chromatin. The authors want to carefully discuss this or perform additional experiments to eliminate indirect/non-specific interactions through NCPs/chromatin.

---

## [Author Response]

Summary:The authors provide a combination of CRISPR/Cas9 knockout, co-immunoprecipitation, and ChIP-seq studies to argue that CHD4 is a potential therapeutic target for the treatment of PAX3-FOXO1 fusion-positive rhabdomyosarcoma. This work builds on the previous observation that CHD4 knockout decreases the growth of FP-RMS cell lines. The studies are fairly thorough and most of the experiments appear well-done. However, the primary conclusions that CHD4 functions independently of NuRD and that there exists a NuRD-only complex are not well supported due to several flaws in the approach and interpretation.

We would like to express our gratitude to the reviewers and the Senior Editor of *eLife* Dr. Kevin Struhl for their interest in our study as well as their very thorough and thoughtful comments, which have helped us to considerably improve our manuscript. We have addressed all raised concerns by adjusting the text and providing novel experimental data as requested. Most importantly, as suggested, we provide novel ChIP-seq data in other FP-RMS cell lines as well as for MTA2 in RH4 cells and performed double knockouts of the NuRD orthologs to support our claims. Our point-by-point responses to the reviewers’ comments are detailed below:

Essential revisions:1) Only one cell line was used throughout the entire mechanistic study. Are there other PF-positive cell lines that the authors could use to validate at least some of the findings?

Indeed, only one cell line was used to explore the mechanism by which CHD4 co-regulates PAX3-FOXO1-mediated gene expression. Therefore, as suggested by the reviewers we have now validated our findings in other fusion-positive rhabdomyosarcoma (FP-RMS) cell lines. First, we investigated if the results of our NuRD-centered CRISPR screen performed in RH4 cells are valid across other FP-RMS cells. To do so, we took advantage of the publicly available CRISPR-based genome-wide cancer vulnerability screen (CRISPR Avana Public 19Q2, depmap.org) and analyzed 5 other FP-RMS cell lines (RH28, RHJT, CW9019, JR, and RH30) as well as RH4 cells for their sensitivity to knockouts of NuRD subunits. This analysis confirmed that FP-RMS is highly dependent on CHD4 and RBBP4 for tumor cell proliferation (Figure 1—figure supplement 1E).

For our finding that in FP-RMS CHD4/NuRD binds to enhancers and super-enhancers together with the tumor driver PAX3-FOXO1 and that NuRD binds more frequently to promoters without CHD4 in RH4 cells, we expanded the experiments to two other FP-RMS cell lines, RH5 and SCMC cells. We performed ChIP-seq assays on these cell lines for PAX3-FOXO1 (using a breakpoint-specific antibody), CHD4, HDAC2, RBBP4, and MTA2. These experiments confirmed that in at least two other FP-RMS cell lines CHD4/NuRD is present at enhancers and super-enhancers and co-localizes with the fusion protein PAX3-FOXO1 in roughly 50% of the fusion protein-specific binding sites (Figure 3—figure supplement 1A and Figure 4—figure supplement 2). Also, we observed that, in SCMC cells, NuRD locations without CHD4 are more frequently found in promoters and in the vicinity of TSSs (Figure 3—figure supplement 1B), although in RH5 cells such difference was less clear.

2) The authors perform a CRISPR/Cas9 screen of major NuRD components which shows that knockout of CHD4 and RBBP4 reduce viability of FP-RMS cells to a greater extent than other components. Based on this observation, they conclude that CHD4 functions independently of the full NuRD in FP-RMS (subsection “CHD4, unlike other NuRD members, is essential for FP-RMS cell viability”, first paragraph). However, there are multiple orthologs for all NuRD core components that may or may not functionally substitute for one another. As they demonstrate in the supplementary material, multiple orthologs are expressed in these cells (MBD2 and MBD3, MTA1 and MTA2). Therefore, although knockout MBD2 or MBD3 alone does not recapitulate the phenotype of CHD4 knockout, simultaneous knockout of both MBD2 and MBD3 indeed recapitulates the phenotype (personal communications). Hence, the conclusion that CHD4 is acting independently of NuRD is not correct. The authors need to either modify this conclusion or provide additional data by simultaneously eliminating functionally substituting orthologs (e.g. MBD2/MBD3 or MTA1/MTA2/MTA3).

The reviewers raise an important point here. In fact, some of the NuRD subunits may have redundant functions which makes the single knockout of one paralog insufficient to interfere with the activity of the subunit, as has been shown for the HDAC subunits (Jurkin et al., 2011). However, MBD2/3 subunits are mutually exclusive subunits of NuRD, and they form distinct NuRD complexes with different functions (Le Guezennec et al., Molecular Cellular Biology, 2006). Hence, in a first approach, we made single knockouts of these subunits to eliminate MBD2- or MBD3-containing NuRD complexes. Nevertheless, eliminating MBD2-contaning NuRD complexes allows the formation of MBD3/NuRD. Therefore, as suggested by the reviewers, we performed double knockouts of MBD2/3 as well as of HDAC1/2, and GATAD2A/B (Figure 1—figure supplement 1F) and observed a consistent decrease in FP-RMS cell proliferation for all double knockouts although less pronounced than the one observed for CHD4 single knockout (median of KO/Control ratio obtained with the 5 sgRNAs tested: CHD4 – 51%; HDAC1/2 – 59%; GATAD2A/B – 65%; MBD2/3 – 68%). Thus, we changed our claim that CHD4 acts independently of NuRD to: “FP-RMS is particularly sensitive to CHD4 depletion amongst all NuRD subunits”.

3) The authors claim that a NuRD-only complex localizes to distinct regions (TSS) as compared to CHD4-NuRD. This claim is based on ChIP-seq of CHD4, HDAC2, and RBBP4 (subsection “CHD4/NuRD localizes to enhancers while CHD4-free NuRD to promoters”, last paragraph). However, as the authors later acknowledge (Discussion, second paragraph) RBBP4 and HDAC2 proteins are found in other chromatin-associated complexes (e.g. RBBP4 is found in PRC2, NuRF, and SIN3 complexes; HDAC2 is found in SIN3 and CoREST complexes). Hence, ChIP-seq of HDAC2 and RBBP4 does not necessarily reflect a NuRD-only complex and is inappropriate for this analysis and to make this claim. ChIP-seq of the MTA proteins would be much more appropriate for this experiment. As far as I am aware, the MTA proteins have not been found in other chromatin complexes; hence, they function as core NuRD components (Zhang et al., 2016) Therefore, the authors need to perform ChIP-seq for MTA proteins instead of HDAC2/RBBP4 in order to support this conclusion.

Since RBBP4 and HDAC2 are present in other complexes besides NuRD, we agree with the reviewers that the overlap of RBBP4 and HDAC2 ChIP-seq peaks might not only reflect the location of NuRD in the genome. On the other hand, as pointed out by the reviewers, the MTA subunits are, to our knowledge, specific to the NuRD complex (Basta and Rauchman, Translational Research, 2015). Therefore, we followed the suggestion of the reviewers and acquired ChIP-seq data for MTA2 in RH4 cells as well as reformulated our analysis depicted in Figure 3, Figure 3—figure supplement 2, Figure 4, Figure 4—figure supplement 1 and 3. In our new analysis, we define NuRD complex locations as the overlap between HDAC2, RBBP4 *and* MTA2 ChIP-seq signal. These newly defined NuRD locations confirm our previous claim that NuRD together with CHD4 is mainly located at enhancers while in the absence of the remodeler it localizes more frequently to promoters and closer to TSSs (Figure 3).

4) The authors endogenously Flag-tag the CHD4 protein to identify co-purifying proteins by mass spectrometry analyses. Based on this work, they develop a model in which CHD4 interacts with a different set of chromatin-associated proteins (BRD4). However, the key challenge with this approach is that CHD4 (NuRD) strongly interacts with chromatin. Hence, it is very difficult to determine whether co-purification reflects direct protein-protein interaction or indirect binding bridged by chromatin. While CHD4 itself has not been shown to directly bind DNA (as the authors point out), it does have chromatin-binding domains (PHD and chromodomains). Of note, histone proteins are among those identified in the mass-spectrometry data (Figure 2) indicating that NCPs are being co-purified. Furthermore, most of the co-purified proteins are from other chromatin-associated complexes or NuRD. These results suggest to me that the majority of the co-purified proteins could easily be explained by indirect/non-specific interaction through NCPs/chromatin. The authors want to carefully discuss this or perform additional experiments to eliminate indirect/non-specific interactions through NCPs/chromatin.

Affinity purification followed by mass spectrometry (AP-MS) is a sensitive and selective method to characterize protein-protein interactions that has been widely used to describe interactions of chromatin regulators and transcription factors (Lambert et al., Journal of Proteomics, 2014; Li X et al., Molecular Systems Biology, 2015, DOI: 10.15252/msb.20145504). However, we agree with the reviewers that using AP-MS to identify the interactome of CHD4 can also lead to the identification of non-specific chromatin/DNA-mediated interactions. To prevent the identification of indirect CHD4 interactions, we used endogenously flag tagged CHD4 for our flag pull downs instead of ectopically overexpressed CHD4 and performed our interactome study in the presence of the endonuclease benzonase (Author response image 1 and subsection “CHD4 interacts with negative and positive regulators of gene expression including BRD4”) to lower the potential identification of interactions mediated by long DNA stretches. Incorporating nucleases in AP-MS has been successfully applied by others who have shown that nuclease digestion allows the identification of both soluble and chromatin-bound interactors of chromatin regulators as well as it eliminates most non-specific interactions mediated by DNA (Li X et al., Molecular Systems Biology, 2015).

Importantly, since nuclease wildtype N-Flag C-Flag digestion increases chromatin solubility, we have also performed CHD4 interactome studies in the absence of benzonase (data not shown), which mitigates the identification of chromatin-associated protein complexes, and we were still able to co-purify all NuRD members (HDAC1/2, MTA1/2/3, MBD2/3, RBBP4/7, GATAD2A/B, CDK2AP1), SWI/SNF subunits (BCL11A, BCL11B, SMARCA4) and other chromatin regulators such as BRD3 and EHMT1, but no histones. However, we agree that it is possible that some of the proteins co-purified with CHD4 could result from residual co-purified chromatin. Hence, as proposed by the reviewers, we have further critically discussed this in the subsection “CHD4 interacts with negative and positive regulators of gene expression including BRD4”.

Finally, we thank the reviewers for their detailed critique which has helped us to considerably improve the manuscript. We trust we have addressed all questions and the manuscript can now be found acceptable for publication.

**Author response image 1. respfig1:** Benzonase digestion of RH4 nuclear extracts used for AP-MS.